# Wearables in Chronomedicine and Interpretation of Circadian Health

**DOI:** 10.3390/diagnostics15030327

**Published:** 2025-01-30

**Authors:** Denis Gubin, Dietmar Weinert, Oliver Stefani, Kuniaki Otsuka, Mikhail Borisenkov, Germaine Cornelissen

**Affiliations:** 1Department of Biology, Tyumen Medical University, 625023 Tyumen, Russia; 2Laboratory for Chronobiology and Chronomedicine, Research Institute of Biomedicine and Biomedical Technologies, Tyumen Medical University, 625023 Tyumen, Russia; 3Tyumen Cardiology Research Center, Tomsk National Research Medical Center, Russian Academy of Sciences, 634009 Tomsk, Russia; 4Institute of Biology/Zoology, Martin Luther University, 06108 Halle-Wittenberg, Germany; dietmar.weinert@zoologie.uni-halle.de; 5Department Engineering and Architecture, Institute of Building Technology and Energy, Lucerne University of Applied Sciences and Arts, 6048 Horw, Switzerland; oliver.stefani@hslu.ch; 6Tokyo Women’s Medical University, Tokyo 162-8666, Japan; frtotk99@ba2.so-net.ne.jp; 7Halberg Chronobiology Center, University of Minnesota, Minneapolis, MN 55455, USA; corne001@umn.edu; 8Department of Molecular Immunology and Biotechnology, Institute of Physiology of Komi Science Centre, Ural Branch of the Russian Academy of Sciences, 167982 Syktyvkar, Russia; borisenkov@physiol.komisc.ru

**Keywords:** wearables, actigraphy, circadian rhythms, health, monitoring, chronobiology, neurodegeneration, metabolism, risk factors, epidemiology, longevity, biological age

## Abstract

Wearable devices have gained increasing attention for use in multifunctional applications related to health monitoring, particularly in research of the circadian rhythms of cognitive functions and metabolic processes. In this comprehensive review, we encompass how wearables can be used to study circadian rhythms in health and disease. We highlight the importance of these rhythms as markers of health and well-being and as potential predictors for health outcomes. We focus on the use of wearable technologies in sleep research, circadian medicine, and chronomedicine beyond the circadian domain and emphasize actigraphy as a validated tool for monitoring sleep, activity, and light exposure. We discuss various mathematical methods currently used to analyze actigraphic data, such as parametric and non-parametric approaches, linear, non-linear, and neural network-based methods applied to quantify circadian and non-circadian variability. We also introduce novel actigraphy-derived markers, which can be used as personalized proxies of health status, assisting in discriminating between health and disease, offering insights into neurobehavioral and metabolic status. We discuss how lifestyle factors such as physical activity and light exposure can modulate brain functions and metabolic health. We emphasize the importance of establishing reference standards for actigraphic measures to further refine data interpretation and improve clinical and research outcomes. The review calls for further research to refine existing tools and methods, deepen our understanding of circadian health, and develop personalized healthcare strategies.

## 1. Introduction

Rhythms are recognized as hallmarks of health [1]. These oscillations can be measured and quantified utilizing diverse mathematical methods and modeling approaches. When such oscillations deviate from their optimal ranges, they can be harnessed as early warning signs for the detection of impending risk and for the implementation of countermeasures to improve health outcomes [2,3,4,5,6,7], epitomizing a pivotal element in the burgeoning domain of circadian medicine [4,7,8,9]. Perturbations in circadian rhythms, both at the molecular and systemic levels, are correlated with the processes of aging and the pathogenesis of diseases [1,2,3,7,10,11,12]. The meticulous assessment of the normal range of physiological variables necessitates the employment of sophisticated monitoring techniques that involve both data collection and data analysis for a chronobiological interpretation of the results. In the field of circadian medicine research, and chronomedicine more generally (also concerned with time structures other than circadian rhythms), various wearable technologies are utilized to monitor and study sleep patterns, activity levels, and other physiological markers that align with our biological clocks.

In the investigation of the optimal characteristics of rhythmic structures, quantitative measures are assessed, alongside deviations from the optimum that may be associated with health risks. These deviations typically encompass both parametric and non-parametric measures. Parametric measures are useful for many circadian functions, which can be described using sinusoidal curves or multi-component models. To accurately characterize the parameters of biological rhythms—namely amplitude, phase, and period length—data collection must occur at suitable intervals across one or, preferably, multiple cycles. Ideally, this collection should be sampled at regular, equidistant intervals; however, it is also feasible with irregular intervals [13,14]. Assessing the intrinsic period in real-life scenarios poses challenges, as its determination necessitates specific, controlled conditions within a laboratory setting, such as the “constant routine” or “forced desynchrony protocol” [15]. Conversely, amplitude and phase evaluations are more feasible in real-life circumstances. By collecting data over several cycles, researchers can introduce a “regularity” domain to measure the stability of phase and amplitude across cycles. Non-parametric measures can be preferred for functions with abrupt transition points, such as wake–sleep–wake states, and include characteristics that eschew the sinusoidal rhythm model yet still facilitate the characterization of amplitude, timing, stability, and fragmentation. Longitudinal and dense data sampling enables a thorough assessment of physiological variability and any divergence from the normal range.

Monitoring and wearable technologies offer excellent means to achieve this objective. Physical activity, temperature, heart rate, and light exposure are among the most accessible and trackable variables by wearables. Moreover, these variables hold greater physiological significance and, in addition to mere step counts, allow for the sampling of quantified values. The selection of the most appropriate method for data analysis is contingent upon the specific physiological variable under consideration. For instance, physical activity may be best characterized through non-parametric approaches due to its non-sinusoidal patterns, whereas temperature typically exhibits a more sinusoidal nature. While wearable technologies offer significant advancements in health monitoring, they are not without limitations. For instance, measurements of wrist temperature, though less intrusive, are often marred by substantial noise in comparison to core body temperature. Their timing may be nearly, though not precisely, in antiphase with those of core temperature. They reflect the work of a special mechanism that ensures a decrease in basal temperature in the evening/night due to heat dissipation, which is necessary for falling asleep and maintaining sleep [16,17]. On the other hand, professional-grade wearables, such as actigraphs, facilitate the assessment of phase alignment among various physiological variables. This is particularly pertinent when incorporating measurements of light exposure, which serves as the principal circadian synchronizer.

In the fields of circadian medicine research and chronomedicine, various wearable technologies are utilized to monitor and study sleep patterns, activity levels, and other physiological markers that align with our biological clocks. Some types of wearables already in use or still in development in this area are summarized in Table 1. Overall, innovative wearables and analytical tools work together to derive refined reference ranges and identify new biomarkers to assess health quality and disease risk [18,19,20,21,22,23,24,25]. It has been emphasized that devices should be evaluated for their intended use in relevant populations, such as patients with mild cognitive impairment at risk of developing Alzheimer’s disease for their intended use [26]. The design of new devices should also keep these specific patient groups in mind. Analysis of electrocardiogram time series using machine learning (ML) algorithms made it possible to reliably identify sleep phases, rapid eye movement/non-rapid eye movement (REM/NREM) [27,28], indicating great prospects for using ML in somnology and chronobiology. However, despite the variety of affordable multisensory wearable devices on the market, their wide public use and Bluetooth compatibility, actigraphy remains a tool best validated scientifically for research in chronomedicine. It provides objective, real-life insights into an individual’s sleep, activity, and light exposure patterns, providing full access to raw data, which is highly important in research [20]. Earlier actigraphs’ embedded accelerometers allowed the device to record body movements, providing a representation of activity levels over extended durations [29,30,31]. It was commonly used to assess sleep quality but also had applications in various medical fields [32,33], including psychiatry, neurology, oncology [34], and geriatrics. Actigraphy has proven to be a valuable tool in diagnosing and managing various conditions, including depression, narcolepsy [35], Alzheimer’s disease, Parkinson’s disease, and dementia [36,37,38]. Actigraphy also serves as a quantitative measure of physical activity throughout the day. Monitoring movement enables researchers to evaluate an individual’s average activity, periods of sedentary behavior, and compliance with study protocols. To estimate measures of physical activity, including energy expenditure, wrist- and hip-worn devices produce comparable results [39]. Advanced actigraphy units are now equipped with sensors for wrist temperature and ambient light [40], allowing for a more comprehensive examination of rhythms and their perturbations. Through the analysis of activity patterns and wrist temperature, researchers can deduce information about the internal circadian clock and its influence on health.

## 2. Actigraphy Features: Beyond Sleep

Considering the thorough exploration of implications, constraints, and viewpoints of actigraphy within the realm of sleep medicine, as detailed in prior literature [24,41,42,43,44,45], this review aims to encompass the infrequently employed, yet increasingly acknowledged applications of actigraphy for the quantification of physical activity, assessment of metabolic health, and the prognostication of lifespan. Subsequently, this review will also deliberate on the eminent roles of commercially available wearable devices in the sphere of circadian medicine and chronomedicine more broadly, including rhythms with periods outside the circadian domain [1,2,7]. These applications do not solely revolve around sleep, as they also involve quantifying circadian (about 24 h–20 to 28 h range) and extra-circadian (ultradian, with periods less than 20 h, and infradian, with periods longer than 28 h) variability, based on dense (usually every minute) and longitudinal (usually weeklong) sampling. To examine time structures of the rest-activity cycle as it changes over time, actigraphy can be empowered with several mathematical tools. In a parametric approach, the period, τ, is the duration of one cycle; the MESOR (Midline Estimating Statistics Of Rhythm) is a rhythm-adjusted mean; the amplitude is a measure of half the extent of predictable change within a cycle; and the acrophase is a measure of the timing of overall high values recurring in each cycle [13]. It extends from circadian to both ultradian and infradian rhythms in a comprehensive spectral analysis [11,46,47]. Acigraphy also includes a non-parametric approach used to derive the following indices: M10, M10 Onset, L5, L5 Onset, inter-daily stability (IS), intra-daily variability (IV), relative amplitude, circadian function index, etc., as features of physical activity [48,49,50,51,52]. The same parametric and non-parametric approaches can also be applied to the other variables (wrist temperature, ambient light measured at different wavelengths, e.g., red, green, blue, ultraviolet A, ultraviolet B, infrared, or even spectrally resolved). Furthermore, they can be used to calculate novel, more sensitive health markers such as indices of daytime light deficit and nocturnal light excess [53], scalar regression markers [54], phase angles between activity and light [55,56], or integrated variables such as TAP, which incorporates measures of three variables (temperature, activity, and position) [50]. A personalized approach to searching for circadian markers of health and disease, quantified by wearable technologies, can be most fruitful, as certain genetic-, gender-, or geographically related phenotypes may be particularly vulnerable to circadian disruption [53].

Actigraphy acquired a growing interest in studies of metabolic health and circadian syndrome, which can be a more precise definition of metabolic syndrome [57,58]. By continuously tracking physical activity and sleep duration, actigraphy provides valuable data that can be correlated to various metabolic endpoints, such as glucose concentrations, insulin sensitivity, and lipid profiles [56,59,60,61,62,63,64]. This approach offers a holistic view of daily habits and their putative effect on metabolic health. Furthermore, integrating actigraphy data with biobank big data enhances the depth of analysis, allowing researchers to explore correlations between activity patterns, genetic factors, and metabolic outcomes on a large scale. Biobanks, repositories of biological samples and associated health data, play a crucial role in advancing our understanding of complex diseases like diabetes, obesity, and cardiovascular disorders, in addition to neurodegenerative diseases and mood disorders that were extensively investigated previously. Actigraphy is promising in identifying early markers to predict cardio-metabolic risks and providing personalized preventive strategies. Furthermore, actigraphy can be used to evaluate the effectiveness of strategies for improving sleep or optimizing the amount and timing of physical activity.

Several important topics, including the comparative analysis of circuit design, power supply, device longevity, measurement accuracy, noise characteristics, calibration methods, and the advantages and limitations of device types such as GPS, accelerometers, and light sensors, as well as the performance and reliability of wearable devices, warrant specific attention in a separate review, as they are outside our scope herein.

## 3. Overview of Actigraphic Health Markers

Actigraphy and other wearable technologies provide a non-invasive and efficient method for continually monitoring physiological functions in real-world environments. These tools have a wide range of applications in research, clinical practice, and personal health monitoring, making them an invaluable resource for quantifying and comprehending various aspects of health and well-being. They are particularly useful when used in conjunction with the central synchronizer of the circadian clock, light, to measure circadian light hygiene and its influence on health and well-being. By combining objective data on light exposure with information on sleep patterns and circadian rhythms, actigraphy can help to pinpoint strategies for promoting optimal circadian alignment and enhancing overall health outcomes. Most common forms of circadian disruptions that are revealed by actigraphy are summarized in Table 2 and schematized in Figure 1.

A decline in circadian robustness with age and disease is considered a general biological feature and may serve as an unspecific marker of biological age to predict healthy aging and longevity [1,2,10,11,65,66,67,68]. The following parametric/non-parametric indices are commonly used to characterize circadian and 24 h variability of physiological variables such as activity, temperature, and light exposure.

Parametric [13]:y(t) = M + ΣiA_i_cos (2πt/τ_i_ + φ_i_), i = 1, …, k(1)
(usually, τ_1_ = 24 h accounts for most of the variance and its parameters (A_1_, φ_1_) are useful biomarkers)

y—related data; t—time

acrophase (φ_i_), moment in time when component with period τ_i_ reaches its peak in relation to a given reference time;

MESOR—Midline Estimating Statistics Of Rhythm (a rhythm-adjusted mean);

A_i_—Amplitude of component with period τ_i_.

Note that a multi-component cosinor is usually most appropriate (e.g., blood pressure and heart rate are usually fitted with a 2-component model, including the 12 h harmonic term). While composite models yield a better approximation of the rhythm’s waveform, the fundamental component, with a period of 24 h, often accounts for most of the variance, and its parameters can serve as valuable biomarkers.

Non-parametric [48]:

M10—the average of 10 h of highest values.

M10 onset—the start of the M10 period.

L5—the average of 5 h of lowest values.

L5 onset—the start of the L5 period.

RA (relative amplitude); RA = (M10 − L5)/(M10 + L5)

IV (intra-daily variability) estimates how variable activity is within a day and can range from 0 to ∞, where higher values represent higher fragmentation. IV is calculated by the following formula, where *N*—total number of measurements in the full time series, Xi—individual values at time *i*, Xm—mean of all Xi values:(2)$$IV=\frac{N\overset{N}{\underset{i=2}{\sum }}{\left(X_{i}-X_{i-1}\right)}^{2}}{(N-1)\overset{N}{\underset{i=1}{\sum }}{\left(X_{i}-\bar{X}\right)}^{2}}$$

IS (inter-daily stability) measures how constant the rest-activity pattern is between days and ranges from 0 to 1. Values closer to 1 mean more constant rest-activity patterns. Assuming that measurements are binned over hourly equal intervals per day over the whole time series, IS is calculated by the following formula, where *N*—total number of measurements in the overall time series, *p*—number of data points per 24 h, Xi—individual values at time *i*, Xm—mean of the overall time series, Xm_h_—hourly means:(3)$$IS=\frac{N\overset{p}{\underset{h=1}{\sum }}{\left({\bar{X}}_{h}-\bar{X}\right)}^{2}}{p\overset{N}{\underset{i=1}{\sum }}{\left(X_{i}-\bar{X}\right)}^{2}}$$

Note that additional measures of activity-rest patterns were recently proposed [52]: the activity balance index (ABI), transition probability (TP), self-similarity parameter (α), which may have merits beyond IV and IS, being independent of differentiation between rest/activity states that currently lack standardization. However, unlike IV or IS, ABI and α are sensitive to extreme values that could be observed in the data.

The activity balance index (ABI) estimates how balanced activity is during the observation span. It ranges from 0 to 1, with values closer to 1 meaning a more balanced activity distribution.

The transition probability (TP) estimates transitions from rest to activity state (A → R, AR) or vice versa (R → A) at a given time point t. TP ranges from 0 to 1. Higher values indicate higher transitions from activity to rest or vice versa.

The self-similarity parameter (α) approaches self-similarity of the acceleration signal during the observation span. It ranges from 0 to 2. If the values fall between 0 and 1, it indicates that the motion is steady and predictable. When the values are between 1 and 2, it suggests that the motion is more erratic and unpredictable. There are some key points to note: a value of 0.5 represents random noise, 1 indicates a pattern called fractal noise, and 1.5 signifies a random walk, which is a type of movement that seems to take random steps.

The Sleep Regularity Index (SRI) is another novel metric [94,116], which can be applied to evaluate an individual’s sleep patterns for their consistency. The SRI is calculated by determining the percentage probability that an individual’s sleep state (asleep or awake) remains the same when comparing two time points that are 24 h apart. This probability is then averaged over the entire observational period, providing an overall indication of sleep regularity. It can range from 0 to 1. A higher SRI indicates a more consistent sleep pattern, whereas lower values mean greater variability in sleep-wake conditions. SRI can be re-scaled as y = 200 (x − 1/2) to scale in the range from −100 to 100 [94]. After such re-scaling, regularity of sleep patterns gauged by SRI typically ranges between an SRI of 0 (random or irregular sleep) and 100 (structured or periodic sleep).

Different approaches for the precise assessment of circadian phase in field studies can be used with relative merits that may depend on research purposes and conditions [117,118] and also on the extent of alignment between activity and light exposure [119]. For certain purposes, such as delayed sleep–wake phase disorder, light-based methods incorporating multiple linear regressions of light exposure for phase assessment were recommended [88], while activity-based models can be useful in older adults (aged 58 to 86 years) [120] who tend to advance their sleep phase. Activity-based phase modeling has been shown to outperform light-based modeling in predicting the dim light melatonin onset (DLMO) in shift workers [121]. Our own data suggest that the same may hold true for cosinor-based models. However, their effectiveness may depend on ambient light conditions and the circadian light hygiene index [122], both of which can vary significantly with the seasons [123]. Furthermore, in view of substantial individual differences in light sensitivity [124], predictions can vary widely among individuals, influenced not only by genetic factors but also by other co-factors, such as a history of COVID-19. This history can affect circadian patterns and light sensitivity related to actigraphy-based measures [125] as well as blood pressure [126].

Estimates of amplitude and phase as proxies of circadian rhythm may require different methods with distinct inherent strengths in modeling biological oscillations:

Cosinor Analysis was developed specifically for the purpose of modeling cyclic data and is particularly useful for estimating rhythm parameters like amplitude and acrophase [13,125,127,128,129,130,131]. While the single-component model assumes the pattern to be sinusoidal, consideration of additional harmonic terms with periods of e.g., 12, 8, and 6 h can approximate the waveform more precisely, including phase estimation.

Non-parametric Actigraphy Indices [48,49,50,51,52] capture informative features of the circadian rhythms of activity and light exposure, such as their extent of irregularity. However, their precision in quantifying amplitude may be more subjective in interpretation, leading to greater variability in results.

Moving Linear Regression Models used for sleep/wake scoring over short (e.g., 1-min) intervals are widely applied to characterize basic sleep parameters [33,132,133], thereby capturing important features of the circadian rhythm other than the amplitude and phase.

Artificial Neural Networks (ANNs) are capable of handling complex correlations and can therefore determine amplitude and phase for various types of data [134,135,136]. Nevertheless, the efficiency of the networks directly depends on the availability of sufficient data for training. These techniques may not be as easy to use for the purpose of estimating amplitude and phase as is the case with cosinor. Results from a study using actigraphy data of blue light and temperature indicated that the ANN model was capable of predicting circadian timing within ±2 h for most individuals following diurnal schedules. This method, however, did not extend to night shift scenarios [126].

Limit-Cycle Oscillator Modeling effectively represents biological oscillations and can provide information on feedback mechanisms, allowing dynamic estimations of amplitude and phase. On the other hand, their complexity and need for extensive computational resources can limit their practical applications. Limit-cycle oscillator modeling is most effective for estimating parameters with non-sinusoidal patterns and when phase stability is low (shift work) [135,137].

Approximation-Based Least-Squares Methods [33,138] offer significant versatility, being capable of accommodating a wide range of models, including non-linear ones, which makes it particularly useful for estimating amplitude and phase. These methods efficiently reduce residuals to enhance model fit; however, they can require substantial computational resources, especially when working with highly complex models. Additionally, their implementation may be less convenient and more challenging to interpret, potentially complicating their practical application.

Furthermore, several methods can improve cosine decomposition, particularly in investigations of ultradian oscillations and their interpretation:

Fourier transformation decomposes a signal into constituent frequencies, allowing for the analysis of periodic components. It is effective for analyzing periodic signals and identifying dominant frequencies in spectra [139]. Fourier transformation can utilize walking accelerometer data to predict somatic health [140] and can outperform a non-parametric approach in identifying depression [141]. However, like other methods, it can be sensitive to noise, outliers, and non-stationarities of the signal (when mean, variance, and autocorrelation structure are not consistent over time).

Wavelet transformation [142,143] allows a thorough analysis of non-stationary signals and detects transient features in circadian and extra-circadian signals. However, it is more complex to implement and interpret than traditional methods and requires careful and subjective selection of parameters.

Hilbert transformation [144,145,146] is useful for extracting instantaneous frequency and amplitude of oscillations, providing insights into complex rhythmic patterns and their phase relationship [147]. It can be used for analyzing non-linear and non-stationary signals. However, it is also sensitive to noise and requires continuous data; furthermore, interpretation may be challenging without a solid understanding of the underlying mathematics.

The choice between different methods of actigraphy/wearables data analyses largely depends on the specific research goals, the sources of the data, and the expected rhythms to be analyzed. Table 3 provides a brief comparison of the relative merits of the different methods used to analyze data from wearables.

## 4. Interpretation of Deviant Parameters

### 4.1. Faded Circadian Oscillation (Amplitude Decrease)

Examples are reduced relative amplitude of activity with a non-parametric approach: lower daytime physical activity or light exposure (M10) that can be accompanied by higher nocturnal activity or light exposure (L5) is indicative of poorer sleep or light hygiene. Reduced circadian amplitude of activity by cosinor or other non-linear approximations similarly relates to lower daytime but higher evening physical activity, Figure 2. In addition, the regularity of physical activity or light exposure patterns can be assessed by comparing the ratio of the amplitude assessed by separate 24 h cycles (A24) to the amplitude approximated over the entire monitoring period, e.g., week (Aw). When activity or light exposure occurs irregularly, Aw/A24 is reduced. The normalized amplitude (NA) can serve as a valuable metric to quantify both physical activity (PA) and light exposure (LE), Figure 3. It is particularly advantageous for LE, which often assumes extremely low or null values during the nighttime. Applying a log transformation can effectively normalize these values, albeit at the cost of reducing daytime variations when LE is generally high.

Mean values for physical activity, ambient light, and wrist temperature (24 h, diurnal, such as M10, nocturnal, such as L5) can be indicative—and serve as proxies—of health and as mortality and morbidity risk factors for certain diseases (Figure 1, Table 2). Examples are low 24 h physical activity, increased or decreased temperature, and low sunlight exposure. Mean values, however, can be restricted to the daytime or to the nighttime, and this information needs to be specified as it affects its interpretation. Indeed, health hazards stem from lowered diurnal, but elevated nocturnal activity, and from low daylight exposure but light-at-night. Therefore, deviations in circadian endpoints other than the mean value also deserve consideration. Changes in the 24 h amplitude and in circadian robustness are cases in point. They can be estimated by absolute measures of A or RA or their normalized values (relative to the individual mean), NA/NRA. The latter may be preferred when inter-individual variability of averages (or MESORs) is high. Alternatively, a measure of circadian robustness can consider the spectral domain of ultradian and infradian frequencies to incorporate the analysis of the relative amplitudes among particular ultradian and infradian frequencies to the circadian, which adds valuable new information. The circadian rhythm is usually the dominant component with the highest power or amplitude in weeklong records of physiological variables that can be recorded with wearable monitoring devices. The relative decrease in circadian prominence to extra-circadian domains is defined as extra-circadian dissemination, ECD [2,10,11]. Another approach includes integrative values for three related actigraphy variables (temperature, position, and activity, TAP), used to estimate the Circadian Function Index [50]. A decrease in circadian robustness, as gauged by either of these methods, is a feature associated with aging, diseases, and health risks.

To ensure comparability among the aforementioned quantitative indices of circadian health, it is imperative to standardize the following essential sampling attributes across various studies: total duration of recording, frequency of observations within the time series, and the number, timing, and extent of interruptions in the dataset. When data are non-equidistant, a measure of the distribution in time of the observations can help determine the frequency region within which reliable results can be anticipated.

### 4.2. Phase Deviations

Phase deviations—both too late and too early maximal values—may serve as health risk predictors. However, stronger evidence has accumulated to explain a phase delay since it is closely linked to and can be driven by poor circadian light hygiene. Phase relationship—phase lag between light and activity (or sleep and melatonin) that can be predicted by wrist temperature (wT) provides more specific information, which, however, can strongly depend on the power of light signaling that varies profoundly depending on the local light environment and seasonal photoperiodism.

A diminution in circadian robustness may stem from a true reduction in amplitude or a deterioration of phase stability. For example, irregular circadian timing (e.g., irregular sleep–wake patterns), which has been associated with a higher mortality risk [149], leads to a reduced amplitude when data are analyzed over several days but not when data are analyzed over a single day (cycle). Indeed, the most common health risks related to circadian disruption are a reduced amplitude, a deviation from sinusoidality, a delayed phase, a compromised phase alignment, and an increased day-to-day instability and fragmentation of the circadian activity rhythm (Table 2).

Phase instability usually means misalignment between circadian oscillations of the distinct circadian functions. It can be caused by a weakened light signaling due to poor circadian light hygiene [4,5,6,150,151,152,153] or by a compromised light perception due to issues with retinal ganglion cells [69,92,103,154,155]. As far as measurable overt rhythms are concerned, it can also be due to the uncoupling between peripheral oscillations or the perception of central clock signaling to peripheral clocks [7,153,156,157]. Phase scattering may occur without changes in mean phase position, or it can be accompanied by a phase delay or phase advance. Accumulating evidence indicates that a phase delay, which is usually linked to a late chronotype, is a common health risk factor (Table 2). A phase delay is associated with evening light exposure and lack of diurnal light [153], or light perception [154,155], but it can also be attributed to certain clock genes or clock-controlled genes related to chronotype [158].

### 4.3. Fragmentation and Ultradian/Infradian Modifications

Modifications of circadian oscillations may occur within a cycle (fragmentation, or prominence in ultradian harmonics) or between consecutive cycles (scattered, unstable circadian parameters such as the 24 h phase, or higher prominence of infradian components). They can be accompanied by a greater deviation of the best-fitted model from an ideal sine curve, i.e., a reduction in the percentage of rhythm. In actigraphy, the most commonly used indices are the non-parametric indices IV and IS [48], which can be used to predict numerous health hazards (Table 2). Quantified approaches to estimate the loss of regularity can be applied to characterize sleep and its stages [46,47].

In addition to deviations in characteristics of circadian rhythms, modifications can be observed in the relationships between circadian and extra-circadian components, such as regular oscillations in the ultradian (periods shorter than 20 h) or infradian (periods longer than 28 h) domains. Regular movement patterns during sleep (locomotor inactivity during sleep) were first described by pioneers in sleep research [159]. They were observed using early accelerometer-based techniques [30] and again recently in modern actigraphy studies [46]. Regular ultradian oscillations with a period of about 110 min characterize activity during sleep. Such oscillations gradually fade across the night. The amplitude of this ultradian rhythm does not depend on sex but is modified by sleep duration and shift work, and gradually declines with age [46]. Ultradian sleep cycles may belong to endogenous biological rhythms and are influenced by factors such as age, gender, and the balance of sleep needs; however, in individuals who are healthy and sleep well, these cycles do not directly change in response to moderate environmental stimuli [47]. Infradian rhythms can be related to geophysical and social cycles and include yearly, monthly, and weekly fluctuations [160,161,162,163]. There is indirect evidence that circaseptan (about-weekly) rhythms [164,165] and circa-lunar cycles [166,167,168] may also be partly endogenous.

In addition to non-parametric indices, inter-daily stability and intra-daily variability can also be assessed by using an ECD model, such as the computation of ultradian-to-circadian and/or infradian-to-circadian rhythm amplitude ratios [2,10,11]. The relative prominence of circadian, ultradian, and infradian rhythms changes with age and disease for different physiological variables such as heart rate, blood pressure, and temperature [11,12,66]. Due to the fact that changes in ultradian and infradian components are influenced by distinct physiological mechanisms [2,11], it is essential to utilize these methods to examine variability and identify disease-specific predictors. As modern wearables acquire a wider range of functions, which can be naturally more sinusoidal than physical activity, characterization of spectral domains (ultradian, circadian, infradian) and their interplay can become a fruitful approach.

Increased sleep irregularity gauged by SRI was linked to several health hazards: a higher 10-year risk of cardiovascular disease, as well as higher rates of obesity, hypertension, elevated fasting glucose levels, hemoglobin A1C, and diabetes [169]. Additionally, higher SRI was associated with higher levels of perceived stress and depression [169,170] and higher body mass index [171]. SRI also showed a U-shaped relationship with the risk of developing dementia. Irregular sleep patterns could be a new risk factor for dementia, and gray matter and hippocampal volume [172].

### 4.4. Misalignment (Intrinsic Desynchrony)

Increased intrinsic desynchrony or circadian misalignment between variables has been shown to be a significant predictor of health risks (Table 2). Examples include the light-activity phase lag [56] and the surrogate index of phase of entrainment, specifically the sleep–temperature phase relationship [173,174]. The temperature phase can serve as a substitute for the golden standard of the melatonin phase, as the phases of temperature and melatonin exhibit a relatively close relationship [175,176,177]. Misalignment between the central clock phase and phases of local or peripheral oscillations [103] can be assessed by the integration of novel sensors into wearables.

### 4.5. Social Jet Lag: Objective Characterization by Wearables

Social jet lag (SJL) refers to the regular misalignment between the personal endogenous circadian clock and social obligations, such as work or school schedules [178,179]. It can occur when individuals stay up later and sleep in on weekends or holidays, causing their internal body clock to be out of sync with their typical weekday routine. It can lead to fatigue, difficulty concentrating, and overall reduced well-being [179]. Greater SJL is associated with an increased risk of cardio-metabolic [107,108,109,110,111] and mood [110,112,113,114] disorders. As actigraphy provides objective measures of activity and temperature, these data can be used to assess circadian health and phase changes on a daily basis in weeklong records [180]. The most abundant actigraphy data that were collected and made available for large database analysis include physical activity. When adopting sampling frameworks that account for weekly patterns, it becomes crucial to calibrate or rectify the data to reflect the actual number of workdays versus days off, including instances of shift work or sporadic commitments that necessitate awakening outside the regular schedule. Furthermore, incorporating details regarding the timing and length of daytime naps enriches the dataset, particularly for the accurate calculation of sleep-related metrics. In older adults, SJL very often approaches zero, which is due to the fact that upon retirement, the main external factor (work schedule) ceases to affect them [111]. Therefore, studying the acute effects of SJL on the health of older people is usually ineffective. The most promising in this case is the study of the chronic effects of SJL on human health.

### 4.6. Composite Markers

Searching for markers that are specific to the risks of certain pathologies can yield positive results when analytical focus is placed on timeframes that correlate with phases of increased sensitivity and physiological response (Table 2). Examples are: (1) areas above (nocturnal excess index, NEI) or under (Daylight Deficit Index, DDI) the curve of optimal 24 h exposure to light [53], where thresholds are based on existing consensus recommendations [181]; (2) MLiT500, the average clock time of all aggregated data above 500 lux [104]; and (3) FOSR (function-on-scalar regression), a method seeking time-of-day differences in a comparison of models fitted to groups defined by the presence or absence of a given risk factor or pathology [54]. NEI and MLiT500 were predictive of body mass index; NEI was also linked to metabolic issues within the 9:30 to 00:30 timeframe, specifically in carriers of common melatonin receptor gene polymorphism, MTNR1b rs10830963; DDI was linked to elevated cortisol concentrations [53]. The FOSR method detected daytime and evening timeframes of higher activity and variability in activity, which were predictive of Alzheimer’s disease with PET-confirmed beta-amyloid Aβ deposition [54]. Novel wearable-based digital markers, encompassing circadian amplitude, phase, and physical activity for biological age evaluation [131], can be useful for further advancement in preventive strategies of circadian medicine.

In relation to blood pressure and heart rate, chronobiological reference values were derived to assess the percentage time elevation/reduction, the amount of excess/deficit, and their timing along the 24 h scale. Similar reference values, qualified by gender and age, derived for all circadian parameters, identified new risk factors related to the amplitude and phase. These abnormal features of the circadian variation in blood pressure and heart rate, known as vascular variability disorders (VVDs), were shown to predict adverse cardiovascular outcomes in several outcome studies [84,85,182,183].

## 5. Circadian Health Markers from Actigraphy

As major markers of circadian disruption described in Table 2 are unspecific and can be related to frailty, aging, or different diseases, further study should seek more specific manifestations of compromised circadian robustness. They can be successful in tracking specific variables and specific indices, also seeking polymorphic gene variants in core-clock or clock-controlled genes [7]. Several genes (e.g., CLOCK, BMAL1, PER1,2,3, and MTNR1B) have SNPs that were consistently linked to compromised circadian health related to cardio-metabolic [5,53,184] or neurodegenerative [5,120,121,184,185,186] diseases.

Establishing reference standards for actigraphy-derived measures is a pressing need that has begun to be addressed. These standards are crucial for providing benchmarks or normal ranges against which individuals’ sleep–wake patterns, activity patterns, and circadian and light hygiene patterns can be compared. By defining these norms, we can better understand deviations that may indicate underlying health issues, facilitating more accurate diagnoses and personalized treatment strategies. Developing reference standards for actigraphy-based parametric and non-parametric indices is a significant step forward in the field of circadian medicine and sleep research. Establishing reference standards for different indices of the distinct measurable variables will allow better interpretation of wearable-collected data and facilitate comparisons across studies and populations. It will enable clinicians and researchers to identify deviations from normal ranges, which may indicate sleep disorders, circadian rhythm disturbances, or other health issues. When light exposure is tracked, drastic seasonal changes, especially at higher latitudes, must be considered as well [53]. Certain individual phenotypic traits, such as disease conditions [26] or skin color [20], that may have an effect on data quality should also be considered.

Recent studies have delved into deviations from optimal circadian health using actigraphy, with a significant portion of them utilizing extensive databases like the UK Biobank [36,70,71,72,73] and NHANES in the USA [68,74,75,76]. While many studies focused on disruptions in the circadian rhythm of physical activity, some also explored deviations in temperature and light exposure rhythms. The following sections outline key findings that highlight the associations between actigraphy or wearable-based circadian measures of health and various health outcomes, including morbidities, frailty, mortality, and longevity. Table 2 provides a comprehensive summary of these studies, detailing the specific aspects of circadian health that were found to be altered in relation to health status. These aspects, however, can strongly depend on the power of light signaling that varies profoundly depending on the local light environment and seasonal photoperiodism.

The circadian rhythm of physical activity plays a crucial role in maintaining bodily [89,187,188,189] and mental [90,190,191] health. A recent retrospective assessment of actigraphy data from a large UK biobank cohort (over 100,000 participants, 40–69 years of age) revealed that a large amplitude of the circadian activity rhythm is associated with lower risks of numerous health issues, including cardiovascular, metabolic, respiratory, infectious, cancer, and all-cause mortality [71]. Similar results were obtained in a prospective 16-year-long survey of over 1000 adults [72]. Moreover, 73 out of 423 (17%) disease phenotypes were significantly associated with a reduced circadian amplitude of the wrist temperature rhythm in actigraphy records from over 100,000 participants, the strongest associations found in relation to type 2 diabetes, non-alcoholic fatty liver disease, hypertension, and pneumonia [73]. A smaller amplitude and reduced circadian robustness were associated with an increased susceptibility to mood disorders such as major depressive disorder and bipolar disorder, mood instability, and neuroticism [70]. Similar markers predicted cognitive deficits in Alzheimer’s and Parkinson’s diseases [36,38]. A larger RA and lesser fragmentation (higher IV, lower IS) of wrist temperature were associated with better sleep quality in Japanese adults [77].

In addition to amplitude and fragmentation, a deviant circadian phase or deviant non-parametric estimates of indices such as M10 or L5 Onset were also indicative of elevated morbidity/mortality risks. Besides a smaller amplitude, a delayed activity phase was related to higher mortality from cancer and stroke [187] and higher risks of dementia or mild cognitive impairment in community-dwelling elderly women [190]. Earlier or later L5 onset vs. intermediate values in the range of 3:00–3:29 from 7-day actigraphy records from 88,282 adults in the UK Biobank was linked to a 20% higher risk of all-cause mortality [91]. Another study including records from 103,712 UK Biobank participants found a relationship between earlier or later sleep onset (compared to optimal onset between 22:00 and 23:00) and the risk of developing cardiovascular diseases, particularly in women [93].

Given that L5 onset and sleep mid-time are markers of chronotype, both studies suggest that the relationship between chronotype and health hazards is non-linear, as opposed to linear. Since chronotype changes with age (also in the 40–70 age range) sex-dependently [192,193], as shown in the UK Biobank database, a more sophisticated analysis is warranted that adjusts some metrics from the UK Biobank or other large databases for age and gender. Such adjustments in the estimation of circadian timing can be helpful to address the question of whether the optimal circadian phase position can be assessed considering the circadian resonance concept, which links longevity to an intermediate morning chronotype. This concept is expected to be indicative of the precise intrinsic circadian clock phenotype in humans [158,194].

As was first revealed by self-measurements for body temperature, amplitude and phase disruptions are present in prediabetes and further worsen in diabetes [78]. Actigraphy validates and further develops approaches to quantify metabolic risks related to obesity and diabetes [53,56,61,62,63,79,195,196]. Again, a smaller circadian amplitude and a delayed phase [62], a weaker coupling between light and activity [56], and a reduced quality of circadian light hygiene, such as excessive evening light [53,63], are linked to greater metabolic risks. In a study of 84,790 UK Biobank participants wearing light sensors for a week, the circadian amplitude and phase were modeled from light data, showing a relationship between exposure to light at night and type 2 diabetes risk [79]. A higher risk of type 2 diabetes was associated with brighter night light exposure, a smaller circadian amplitude, and a displaced position of the circadian phase, independently of genetic risk factors [79]. In this study, an association was also found between a higher type 2 diabetes risk and lower daytime light levels, which became non-significant when physical activity was included as a co-factor in the model.

Given the interplay between physical activity and daylight on the circadian amplitude and robustness [197], it also suggests that physical activity may influence the relationship between light exposure and diabetes risk. However, another mechanism for the association between circadian misalignment and metabolic disorders cannot be ruled out. Zambrano et al. [198] showed that a simultaneous increase in insulin and melatonin levels leads to a deterioration in insulin signal transduction in adipocytes in vitro. It cannot be excluded that a similar effect will be caused by regular consumption of high-glycemic-index food late in the evening/at night, as well as regular intake of pharmacological preparations of melatonin before dinner. Overall, aligned circadian rhythms of physical activity and daylight exposure, with sufficient activity and light exposure during daytime hours, facilitate a strong circadian amplitude and robustness, also of the circadian rhythm in temperature [197,199,200,201,202]. Small amplitudes and shifted phases of circadian rhythms in physical activity are attributes of depressed individuals [80,81,82] and of individuals with binge eating disorder [83]. Fragmentation of the physical activity rhythm was linked to food addiction and emotional eating in young adults [203].

## 6. Molecular Insights on the Interaction Between Timed Physical Activity and Brain Health

Regular physical activity and optimally scheduled exercise exert multifaceted effects on health and well-being, helping to improve metabolic health via modulating the lipid profile [204,205,206,207] and reducing inflammation [208]. Optimization of exercise by its timing can be particularly useful for maintaining healthy triglycerides, TG [209], and the ratio between TG and high-density lipids, HDL, since these lipids are closely coupled with circadian factors [53,210,211]. Effects of optimally scheduled physical activity involve the modulation of metabolomics [212], which establishes the molecular background for brain health [208,213]. Even short-term (about 30 min) bouts of physical activity improved inflammatory profiles by increasing adiponectin while decreasing leptin and interleukin-6 [214], while also increasing brain-derived neurotrophic factor (BDNF) and insulin-like growth factor 1 (IGF-1) [215]. Regular physical activity can influence protein synthesis and breakdown, which can affect the 24 h balance of lipids and amino acids. For instance, scheduled physical activity can provide cardio-metabolic protection by stimulating the production of lactoylphenylalanine, or Lac-Phe, by carnosine dipeptidase II (CNDP2) cells in diverse tissues that reduces food intake without affecting motor activity or energy expenditure [216]. It has potential effects on food addiction, mood, depression, and anxiety, and it also regulates the timing of food intake, which can be a necessary co-factor for exercise to maintain metabolic health [217,218].

Acute moderate physical exercise improves lipid metabolism via increasing circulating 12,13-dihydroxy-9Z-octadecenoic acid (12,13-diHOME) in men and women of different age groups [219]. Activity modulates regulatory factors of the circadian rhythm (Dbp, Tef, Nr1d2, and Per3) in mesenchymal stem cells, the process that was shown to be crucial for multi-tissue molecular responses to exercise and obesity [220]. Physical activity is coupled with the central carbon metabolism, which involves glycolysis and the pentose-phosphate pathway (PPP) that works to oxidize glucose, resulting in the production of NADH and NADPH. NADPH metabolism modulates circadian rhythm parameters, including the period length of the circadian activity rhythm via the pentose-phosphate pathway [221].

Such interaction between the circadian clock and the timing of physical activity recruits the nuclear factor Nrf2 (erythroid-derived 2)-like 2) pathway. Nrf2 is a transcription factor involved in the expression of over 250 genes, i.e., providing feedback between redox oscillations and the circadian transcriptional rhythms through the secondary core-clock loop gene NR1D [221,222]. Nrf2 also affects brain functions, increasing glucose uptake in neurons and astrocytes [223]. Nrf2 is also involved in maintaining large-amplitude circadian rhythms in organs and tissues by regulating expression of the core-clock Cry2 gene [224]. Therefore, major beneficial effects of timed exercise may rely on the interaction between Nrf2 and glucose metabolism [225].

Greater benefits of evening exercise in patients with metabolic disorders are hypothesized to be explained by the Nrf2-dependent activation of the antioxidant response element (ARE) region in skeletal muscles, binding to the promoter region of the interleukin 6 (IL-6) gene. Transcription and translation of IL-6 protein are thereby increased, which then stimulates AMP-activated protein kinase (AMPK), which itself is activated by exercise and also activates Nrf2, closing this feedback loop [225]. Physical activity activates PPARs (peroxisome proliferator-activated receptors) [226] that help to integrate the circadian clocks with energy metabolism [227] and are expressed in mammalian tissues in a circadian-dependent manner [228]. Given that the balance between these factors differs between morning and evening, the effects of exercise at different times of day also vary [225].

In a mouse model, lifelong exercise exerted geroprotective effects by restoring (mainly by readjusting) the rhythmic machinery via the core circadian clock protein BMAL1 and resetting circadian transcriptomic programs of younger animals, mainly in the central nervous system and vascular endothelial cells [229]. Sustained physical activity exerts profound effects on the endogenous endocannabinoid system that may help to enhance mood and learning abilities and modulate the endogenous 24 h balance of serotonin, melatonin, and dopamine [230], especially in the presence of sufficient outdoor exposure to sunlight [197].

## 7. Wearables to Track Circadian Markers in Neurodegenerative Diseases

Alterations in the interaction between light exposure and circadian rhythms can be utilized to track early changes associated with neurobehavioral and neurodegenerative pathologies. Precision measurement is essential to increase statistical power and ensure reproducibility in human neuroscience, potentially revolutionizing the field through enhanced methodological rigor and improved scientific outcomes [231]. Research suggests a significant correlation between metabolic changes and the progression of neurodegenerative diseases. Epidemiological studies have consistently linked conditions like obesity and metabolic disorders to the onset of neurodegenerative diseases [232,233]. Hormones such as leptin, ghrelin, insulin, and IGF-1 play a central role in protecting against neuronal damage, resisting harmful stimuli, and other neurodegenerative mechanisms [233].

In addition to actigraphy, wearable GPS sensors and commercial accelerometers offer a non-invasive method for monitoring the mobility and physical activity of individuals with neurodegenerative disorders, potentially serving as biomarkers for disease tracking and treatment response. Despite challenges in standardizing remote monitoring methods [26], recent research [234] has shown limited use of GPS but extensive application of accelerometers in patients with neurodegenerative disorders, resulting in promising clinical trial outcomes and strong correlations with disease progression and patient activities. The next steps involve standardizing these technologies for broader application and validation of their effectiveness in larger patient populations to improve disease monitoring and management. A meta-analysis of 48 studies on circadian disruption in dementia using wearable technology revealed that adults with dementia exhibit lower activity levels, disrupted sleep–wake patterns, increased fragmentation, and reduced normal daytime activity compared to non-dementia individuals [235].

The main circadian brain clock synchronizers, such as light [236,237] and melatonin [92,154,155,238,239,240], can be effective in preventing and/or mitigating neuroinflammation and neurodegeneration. Enhancement of circadian light signaling by daytime light exposure and/or melatonin administration at night can improve circadian robustness by increasing the circadian amplitude, aligning the phase, and strengthening signal robustness in overt phenotypic functions measured by wearables. However, currently, very few studies have addressed light or melatonin therapies using protocols that are personalized for an individual patient’s circadian phase or chronotype. However, a recent phase 2 clinical trial showed that biologically directed daylight therapy improved restorative deep sleep in individuals with mild to moderate Parkinson’s disease, with no significant difference between controlled daylight and melanopsin booster light [37]. The findings suggest that personalized indoor daylight therapy could be effective in improving sleep in early to moderate stages of the disease, calling for further research in advanced Parkinson’s disease.

We recently reviewed newer actigraphs that are available to assess circadian light hygiene for appearance, dimensions, weight, mounting, battery, sensors, features, communication interface, and software [40]. Actigraphs equipped with light sensors can help to track the intensity and duration of light exposure throughout the day, analyze its effect on circadian rhythms, and adjust an individual’s light exposure (circadian light hygiene) according to the recent recommendations [181]. These data help quantify the amount of light a person is exposed to during different times of the day, including natural light from the sun and artificial light sources indoors. The timing of light exposure can be analyzed relative to key circadian markers, such as the onset of melatonin secretion in the evening and the offset of melatonin secretion in the morning. This information helps assess whether individuals’ light exposure aligns with their natural circadian rhythms. By correlating light exposure data from actigraphy with sleep patterns, researchers can evaluate how light exposure influences the timing and quality of sleep.

In a recent study of Arctic residents [53] during the spring equinox, a higher BMI was found to be linked to higher blue light exposure within distinct time windows and also to a lower wrist temperature MESOR. An evening time window between 9:30 p.m. and 0:30 a.m. was identified when 95% confidence intervals of blue light exposure (BLE) were non-overlapping between groups with a BMI < 25 or BMI > 25 kg/m^2^. Such link was coupled to the MTNR1B rs10830963 G-allele. A novel Nocturnal Excess Index (NEI) was suggested to estimate evening light overexposure along with the Daylight Deficit Index (DDI), which was associated with morning cortisol values [53]. Another study used a method known as FOSR to divide an average 24 h activity signal into 48 30 min segments and evaluate differences between two groups within each 30 min segment. This approach found that the Aβ-positive group exhibited significantly higher activity levels than the Aβ-negative group between 1:00 p.m. and 3:30 p.m. Additionally, by analyzing the standard deviation of activity in each segment, the authors demonstrated that the Aβ-positive group displayed more consistent activity patterns during the early afternoon throughout the recording period [54]. In this study, light exposure was not assessed, although it is likely that episodes of elevated activity and corresponding variability could be attributed to simultaneous differences in light exposure. Indeed, some studies suggest that excessive evening light may be linked to cognitive impairment [241,242]. A characteristic motor activity profile in older adults with dementia has been previously described, termed “sundowning”—an increase in the level of motor activity in the late afternoon [234,243,244], occurring in approximately 20% of patients with Alzheimer’s disease [245].

In neurodegenerative diseases, the function of retinal ganglion cells (RGCs), notably the intrinsically photosensitive cells (ipRGCs), is compromised [246,247,248]. Compromised light signaling due to progressive loss of ganglion cells, including ipRGCs, is most evident in advanced stages of glaucoma, being associated with complex circadian disruptions, including changes in circadian alignment and robustness [69,92,103,154,155,211], altered mood [70], and sleep [69,154,155,249]. Similarly, in Alzheimer’s disease, impaired visual function is associated with sleep/wake disorders and cognitive decline [250]. Such alterations can be linked to a decreased amplitude of light–dark alternation, since both melatonin [92,154,155] and daylight [251] mitigate circadian disruptions. Similarly, enhancing spectral lens transmission after cataract replacement improves circadian health [252].

Besides neuroinflammation and neurodegeneration, some other factors, such as a history of SARS-CoV-2 infection, may change the susceptibility to circadian light hygiene, likely affecting light perception or sensitivity. A recent study showed that populations with a history of COVID-19 had lower daytime exposure to white, blue, and ultraviolet B light [253]. Those who had poor circadian light hygiene (estimated by a smaller normalized circadian amplitude) were more susceptible to circadian disruptions, manifested as a phase delay, small amplitude, and less robust circadian patterns of activity and delayed sleep. Furthermore, the longer the time elapsed since their COVID-19 diagnosis, the poorer circadian light hygiene the patients had.

## 8. Boosting Brain, Vascular and Metabolic Health by Clock-Enhancing Strategies

### 8.1. Scheduled Physical Activity

To explore the mechanisms that underpin the relationship between scheduled circadian physical activity and the maintenance of proper light hygiene, we examine their role in promoting strong and stable circadian rhythms. Physical activity is closely linked to circadian health since it occurs regularly within the active phase of the 24 h rhythm, which is usually daytime for humans. The circadian clock can be entrained primarily by light but also by physical activity and meals [197,204,254]. This means that the timing of these daily routines can influence the body’s internal clock, helping to regulate the sleep–wake cycle and other physiological processes. This complex interaction highlights the importance of maintaining consistent routines and healthy habits for overall well-being. Most common health risks related to circadian disruption, such as a reduced amplitude, a delayed phase, increased instability, and fragmentation of rhythms, can be mitigated by physical activity. Animal studies showed the stabilizing effect of physical activity on the circadian rhythm [255,256]. The effect of stable circadian activity rhythms for learning and memory was also shown [257,258].

Exercise timing occurring at the post-absorptive phase was shown to cause greater fat oxidation compared to postprandial exercise, as confirmed by indirect calorimetry and ^13^C magnetic resonance spectroscopy [259]. Timed physical activity, such as morning exercise, decreased abdominal fat and blood pressure in women, while evening exercise improved muscular performance. In men, evening workouts boosted fat oxidation and lowered systolic blood pressure regardless of macronutrient intake [260]. Timed physical activity is promising to handle neurodegenerative pathologies [261].

Furthermore, personalized physical activity adjusted for individual differences [262,263], including individual circadian clock features such as phase and amplitude, may both preserve robust circadian rhythms and enhance benefits of physical activity for systemic and mental health [197]. A study showed that similar to light or melatonin, phase response curves to exercise and physical activity exist [264], assuming that scheduled timing of regular physical activity may facilitate circadian phase correction. A high order of “fitness” or robustness of circadian rhythms is generally linked to a younger age and a better health status [11,12]. Recent UK Biobank-based studies found that engaging in moderate physical activity can help reduce the negative influence of both short and long sleep duration on the risk of all-cause mortality and cardiovascular disease [265] and the effect of short sleep on the incidence of type 2 diabetes [266].

The analysis of UK Biobank actigraphy data from 2324 atrial fibrillation patients revealed that engaging in over 105 min of moderate-to-vigorous physical activity weekly reduces the risks of heart failure and all-cause mortality [267]. Intense sporadic “lifestyle physical activity” was correlated with a reduced incidence of cancer risk, according to a prospective cohort study analyzing 22,398 self-reported sedentary adults from the UK Biobank accelerometry subsample [268]. Increased activity, even when concentrated within 1 to 2 days each week (“weekend-warrior” patterns), may be effective for improving cardiovascular risk profiles [269].

Similarly, both regular and weekend warrior patterns of moderate physical activity, above the 150 min per week recommended by the WHO, are associated with an equal reduction in Parkinson’s disease risk [270]. Another study suggested a linear relationship between physical activity and cardiovascular health, revealing that individuals with higher levels of physical activity consistently exhibit a reduced risk of cardiovascular disease, with the most substantial benefits observed in those who maintain the highest intensity of activity [271]. A longitudinal analysis of 86,556 UK Biobank participants over an average follow-up of 6 years demonstrated that greater overall physical activity and increased daily step count were significantly linked to a reduced incidence of cancers [272]. In a study of 81,717 UK Biobank participants, it was observed that increased engagement in physical activity correlated with reduced hospitalization risks for nine out of the twenty-five most frequent hospitalization causes, with the most significant risk reductions seen in gallbladder disease, diabetes, and urinary tract infections [273]. In a US cohort study involving 7607 adults, an increased accumulation of light-intensity and moderate-to-vigorous-intensity physical activities was found to be inversely associated with stroke risk, whereas extended periods of sedentary behavior correlated with an elevated risk [274].

Our body’s circadian rhythms govern various physiological processes, including the timing and intensity of physical activity. Disruptions to these rhythms, such as irregular sleep patterns or inconsistent exercise schedules, can have a detrimental effect on mental well-being. Vice versa, regular physical activity can contribute to improving mood, reducing stress, and enhancing cognitive functions [275,276,277,278]. Not only aspects of regularity and intensity of exercise were effective to improve health and reduce risks of cardiovascular and metabolic pathologies, but timing of exercise is also important [279,280,281]. Recently accumulated evidence underlines the benefits of exercise scheduled according to the circadian clock and chronotype, also aiding metabolic health [262]. Therefore, optimization of regular circadian timing of physical exercise can be an effective way to manage symptoms of anxiety, depression, and other mental health conditions.

Aligning physical activity with our body’s internal clock can help optimize performance, recovery, and overall psychological well-being. Incorporating exercise into our daily routine at consistent times can help regulate our circadian rhythms, leading to improved sleep quality, increased energy levels, and improved mental clarity. By synchronizing physical activity with our body’s natural rhythms, we can promote psychological resilience, enhance brain function, and better cope with the demands of daily life. To date, personalized recommendations for timing exercise, while important, are usually overlooked. In the case of blood pressure, an N-of-1 study showed that exercise in the evening—but not in the morning—increased the 24 h amplitude. Since the participant had an excessive 24 h amplitude of blood pressure, the personalized recommendation was to exercise in the morning, while for others, evening exercise could be beneficial in strengthening a weakened circadian rhythm [281].

### 8.2. Light Hygiene and Chronobiotics

Circadian health improvement strategies can balance the timing and amount of melatonin and daylight. Boosting circadian signals through exposure to daylight [37,282,283,284,285,286] or administering melatonin during the nighttime [92,287,288] can strengthen the circadian system. Personalization of dose and timing of melatonin administration can be most important for its effective supplementation. Evening low-dose melatonin lowered high blood pressure in hypertensive patients [288] and intraocular pressure in patients with glaucoma [92], with the greatest lowering effect achieved in the morning. Melatonin also reduced glycosylated hemoglobin levels (HbA1c) and increased high-density lipoprotein-cholesterol [97,289]. Melatonin can be more effective in combination with an enhanced dynamic range of light exposure.

Improving the lighting conditions in the surroundings with adaptable lighting systems positively influenced the mood and behavior of elderly individuals [287,290]. In agreement with phase-response curves [291,292,293], morning/daytime light and evening/nocturnal melatonin facilitate circadian phase advancements, preventing complex negative aspects of late chronotypes [294,295,296,297]. The combination of light therapy with melatonin [298,299] or physical activity [300] can be effective in neurodegenerative diseases, including glaucoma [92,154,155]. Light regularity may help to maintain sleep regularity, even in healthy young adults [301].

Timed physical activity may have greater circadian effects when it coincides with daylight exposure [197], better circadian light hygiene per se may have metabolic benefits [122]. At high latitudes, a larger amplitude and an earlier phase of light exposure, mirrored by a greater amplitude and an earlier phase of melatonin and by an earlier sleep phase, characterize seasons with a more comfortable circadian light hygiene. These features are also associated with better proxies of metabolic health, even when there are no differences in the patterns of physical activity [123]. Proper timing of intake of some other substances with pronounced circadian effects, such as coffee, can also provide benefits for metabolism and overall health: morning coffee consumption is strongly associated with lower all-cause, cardiovascular, and cancer-specific mortality [302]. It can also enhance the effects of timed exercise by boosting peak power, readiness for physical efforts, and cognitive performance [303].

Overall, circadian health rescue or enhancement implies maintenance of due amplitude, phase alignment, and stability of the signals that govern overt bodily functions (activity, temperature, heart rate, blood pressure, etc.), trackable by wearable devices.

### 8.3. Optimizing Weekly Schedules

The optimal time to exercise needs to be personalized not only in terms of daytime but also on the weekly scale. Currently, there is a lack of agreement regarding the optimal duration, intensity, timing, and frequency of weekly physical activity for health maintenance. For instance, inconsistencies exist between research findings. One study [269] suggests that one to two weekly sessions of physical activity—termed “weekend warrior” sessions—are adequate for reducing cardiovascular risk comparably to physical activity that is more evenly distributed throughout the week. Other studies, however, conclude that such infrequent activity does not achieve an equivalent reduction in the cardio-metabolic index [304] or lipid accumulation products [305]. A potential explanation for the differing results might be the age of the target population, databases consulted, endpoints used, or the need for individualization. The UK study by Khurshid et al. [269] encompassed individuals aged 40–69 years from the UK Biobank, whereas the research by Xue et al. [304] focused on a distinct cohort from the US National Health and Nutrition Examination Survey (NHANES). The latter study determined that the correlation between physical activity and the cardio-metabolic index was more significant in subgroups aged below 45 or above 60.

An additional crucial detail when studying weekly patterns pertains to the infradian rhythms of physical activity. Employing chronobiological and mathematical modeling to analyze meticulously the distribution of intensity patterns across the 24 h span and throughout the week could provide greater insight into this matter.

## 9. Perspectives of Actigraphy-Compatible Wearable Technologies/Next-Generation Comprehensive Monitoring Systems

The simultaneous tracking of different physiological variables provides a more complete picture of an individual’s circadian health status. The ability to detect subtle changes in physiological and neurological patterns could lead to earlier and more accurate diagnoses of conditions like sleep disorders, depression, and neurodegenerative or metabolic diseases. For research and diagnostic purposes, this integrated approach would require effective methods for combining data from different sources while maintaining data integrity, advanced computational methods to analyze the complex data, ways to extract meaningful patterns, and a strategy to rigorously test the results thus obtained and validate the clinical relevance of the findings.

Focus on extra-circadian frequency domains: ultradian rhythms, or biological rhythms with a period shorter than 24 h, can reflect various physiological and neurological functions. Ultradian fluctuations in brain functions [47,306,307] are under-investigated, though a promising area of research that can utilize ultradian fluctuations and their beat harmonics to untangle complex interactions for a better understanding of biological processes and diagnostics applications. Beat harmonics refer to the composite frequencies produced when two or more ultradian rhythms interact, much like the overlapping waves in acoustics that create a new rhythm or “beat”, providing deeper insights into the synchronization of biological rhythms. Furthermore, in-sync vs. out-of-sync ultradian harmonics may have different outputs, such as non-overlapping lower-frequency oscillations that can be meaningful. The concept of using data from wearables to investigate extra-circadian modulations is innovative and holds potential for advancing medical diagnostics. The analysis of ECG data in specific frequency domains also offers insight into the functioning of specific brain areas [308,309,310,311,312].

### 9.1. Functional Near-Infrared Spectroscopy (fNIRS) and Photoplethysmography (PPG)

For example, integrating variables obtained by photoplethysmography [313,314,315,316,317,318] or functional near-infrared spectroscopy (fNIRS) [319,320] with next-generation wearables may offer a comprehensive approach to monitoring and analyzing physiological and neurological functions. fNIRS stands out as a hemodynamics method that offers significant advantages [321], particularly in terms of its portability and greater motion tolerance, areas where functional magnetic resonance imaging (fMRI) faces constraints [322]. Wearable fNIRS could be applied to better understand time-specified windows of global vs. local brain signal fluctuations and functional connectivity, as suggested by previous fMRI studies [323,324].

### 9.2. Biochemical Analyses

Biochemical analyses are being developed and can be integrated into wearables in the proximal future [20,325,326,327,328,329,330,331]. Sweat-based wearable-enabling technology was recently proposed to track cortisol [332] and inflammation markers such as interleukin-1β and C-reactive protein [326,327]. To advance the understanding of perspiration’s role in thermoregulation and its applied use for physiological monitoring, innovative microfluidic patches with hierarchical superhydrophilic biosponges and integrated sensors should be adopted for efficient, real-time sweat analysis during various activities and conditions [333]. Cutting-edge wearable technologies will have the capability to track the golden standard circadian phase marker, melatonin, but also leptin and ghrelin [20].

### 9.3. Improving Light Exposure Monitoring and Analytics

Melanopic equivalent daylight illuminance (mEDI) is a metric used to quantify the effect of light on the melanopsin-containing intrinsically photosensitive retinal ganglion cells (ipRGCs) in the human eye [181,334,335,336,337]. These cells have a peak sensitivity in the shorter wavelength part of the visible spectrum and play an important role in regulating circadian rhythms. Melanopic EDI measures the effectiveness of light in stimulating these cells, relative to a standard daylight (D65) condition. It is a valuable measure for understanding how different light sources affect our biological clock and other non-image-forming (NIF) effects of light. To accurately determine the melanopic EDI, it is essential to measure irradiance and the spectral distribution of a light source. This is of importance because the color temperature of daylight fluctuates depending on the time of day, and light sources found in various home environments and offices may have different spectral compositions. Further advancements in actigraphy capable of measuring the spectral distribution or directly the melanopic EDI are necessary for this purpose. Furthermore, positioning wearable light loggers or dosimeters closer to the eyes, such as with the lido [338], can refine the assessment of light exposure.

Understanding the user’s challenges of wearing light dosimeters is key to designing devices that are more acceptable and supporting research into light exposure and its effects on health and behavior. To improve the wearer’s experience and compliance, and therefore data quality, the device should be designed to be unobtrusive. Devices should be adapted to the needs of different patient populations and mounting/body location methods. In addition, comprehensive user training and motivational incentives can improve the compliance of the wearer. The implementation of these recommendations will improve the quality of data collected by wearable light loggers. While most actigraph models cannot measure melanopic EDI directly, some are equipped with RGB light sensors [40]. For an overview of current wearable light loggers and their specifications, see [335]. Devices like the ActTrust, for example, can also monitor light intensity in the ultraviolet A (UVA), ultraviolet B (UVB), and infrared (IR) wavelength ranges.

While this feature has not been extensively explored yet, the ability to gather precise data on UV and IR light exposure in real time holds significant potential for various health and therapeutic uses [339,340,341]. For instance, it could be instrumental in managing conditions such as autoimmune diseases, mood and metabolic disorders, as well as addictions [342,343]. Scientifically valid information on UV exposure can also be collected by commercially available wearables [344]. Clinical records underscore the role of circadian rhythms in determining health results and guiding medical procedures, especially when using data from electronic health records. Yet, due to the lack of uniformity in collection methods and equipment, harnessing electronic health records for chronomedicine demands addressing built-in biases and data patterns unrelated to biology [345]. The precision of light logging can be further enhanced by improving light sensors and rendering their position closer to the eyes [338]. Improving their metrology [346] should improve the assessment of physiological effects of natural light and human-centric light technologies [347].

Many commercial light loggers lack transparency in critical technical aspects such as sensor types and data processing methods. This opacity can lead to inaccuracies, undetected errors, or incomplete data sets, making it difficult to ensure the accuracy and comparability of results between different devices. As a result, there is an urgent need for a standardized methodology to collect and analyze data from wearable light loggers. The project “Metrology for Wearable Light Loggers and Optical Radiation Dosimeters” (22NRM05, MeLiDos) will develop the measurement and data analysis methods needed to characterize and validate wearable light loggers. In addition, the JTC 20 (D6/D2) technical report on wearable alpha-optic dosimetry and light logging will provide a comprehensive review of current practices in the scientific literature. Additional sensors incorporating heart rate [348], oxygen saturation [349,350], and functional near-infrared spectroscopy techniques fNIRS [351,352] may enhance the informative capacity of wearable technologies for digital phenotyping in the near future. Despite their limited precision, commercially available smartphone-compatible wearables are already used to evaluate sleep disorders, cardio-metabolic, mental health, and well-being, but the lack of standardization is causing marked disagreement between devices [86,353,354,355,356,357].

### 9.4. Perspectives

Currently, there is consensus on optimal duration of data sampling, inclusion of both free days and working days, minimal sampling requirements, and allowed gaps within samplings. However, there are no clear definitions of normal ranges and how such ranges vary depending on methods of data analysis and critical environmental factors such as local environmental light conditions. Reference standards should aid in the development of diagnostic criteria and treatment guidelines, notably for sleep-related conditions. They would serve as benchmarks for assessing treatment efficacy and monitoring patient progress over time. Overall, the development of reference standards for actigraphy-based indices would enhance the utility of actigraphy in clinical practice and research, ultimately improving our understanding of sleep and circadian rhythms and advancing personalized approaches to healthcare. While deviations from normative values in actigraphy-related variables may often lack specificity, they can nonetheless serve as a catalyst for initiating a comprehensive series of tests. This approach aims to identify the underlying causes, facilitating an earlier diagnosis in the disease progression, when treatment is more likely to be effective.

New sensor technologies, enabling the collection of big data sets, to be organized in large repositories, integrated with artificial intelligence (AI), are expected to further enhance the prediction of health hazards and upgrade the interpretation of circadian health by wearables. Monitoring a broader range of health proxies with standardized sampling requirements, empowered by sophisticated AI algorithms, will allow personalized chronodiagnosis and may encourage large-scale studies to explore the interplay between genetic background, environmental factors, lifestyle factors, and circadian health outcomes. This holistic approach should facilitate the development of targeted interventions to optimize health and well-being in diverse populations, based on their distinct genetic predispositions and specifics of environmental factors and lifestyle. As these technologies evolve, they are expected to offer truly personalized prevention and management capabilities of circadian-related disorders.

## Figures and Tables

**Figure 1 diagnostics-15-00327-f001:**
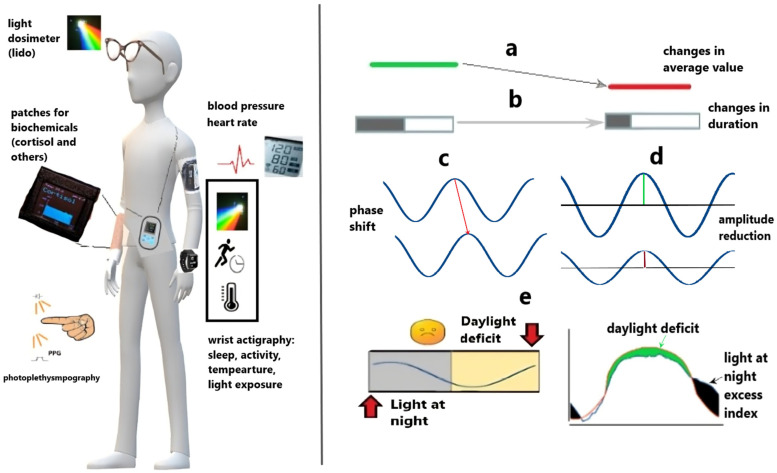
Wearables available and schematics for the most typical circadian deviations. Left: Available monitoring technologies: actigraphy tracked variables (framed): activity, wrist temperature, wrist or hip light exposure; other wearables tracked variables: near-eye light dosimeters (Lidos), blood pressure, heart rate, photoplethysmography (PPG)s, biochemical variables (cortisol, interleukin-1β and C-reactive protein). Right: schematic markers of circadian disruptions: (**a**) deviation of average value; (**b**) changes in the duration of periods of activity, sleep, daylight exposure, etc.; (**c**) phase deviations; (**d**) decrease in amplitude; (**e**) changes in optimal parameters within the time epoch (example of light exposure); suboptimal circadian light hygiene implies deficit of daytime light and (or) excess of light at night, that can be gauged as areas outside the curve of recommendations for optimal 24 h light hygiene; see Figures 2 and 3 for further details).

**Figure 2 diagnostics-15-00327-f002:**
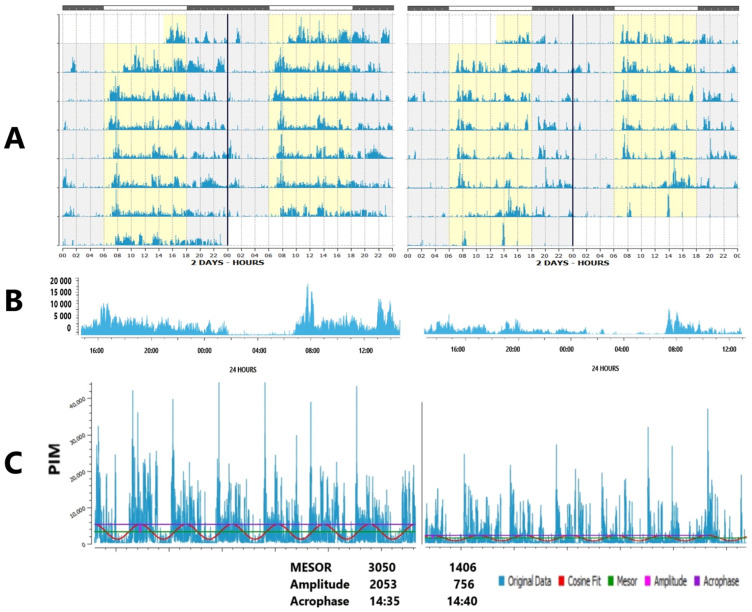
Examples of cases with relatively high (left) and low (right) motor activity with about 2-fold MESOR and almost 3-fold amplitude differences, while acrophases are similar. (**A**) 7-day actograms; (**B**) average 24 h patterns, (**C**) average 7-day patterns with approximated best-fitted 24 h cosine function with its parameters. PIM—activity data processed using proportional integration mode, a measure of activity level or vigor of motion [41].

**Figure 3 diagnostics-15-00327-f003:**
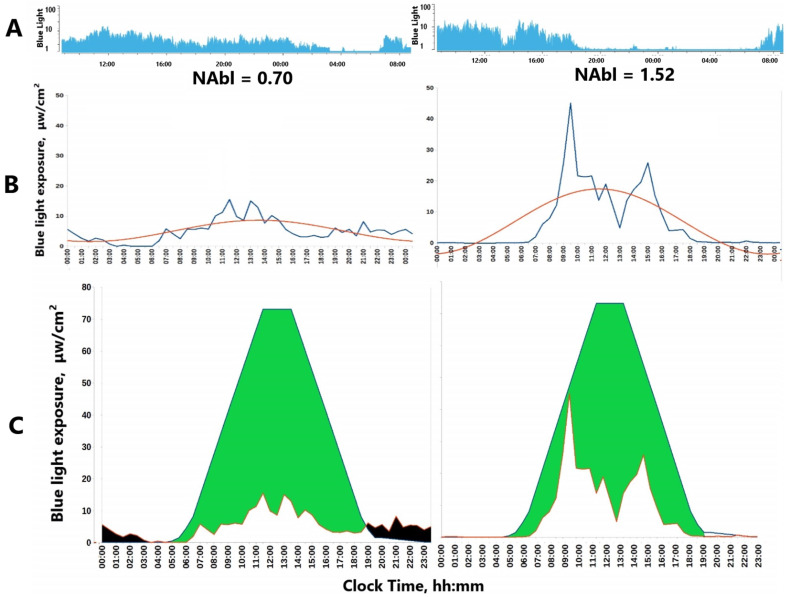
Examples of light hygiene measures: (**A**) 24 h average patterns of blue light exposure in patients with poor (left) and fair (right) circadian light hygiene; (**B**) normalized circadian amplitude of light exposure (NAbl) of individuals with poor (NAbl = 0.7, left) and fair (NAbl = 1.52, right) circadian light hygiene; and (**C**) geometric indices of cumulative daylight deficit and nocturnal light excess [53], our reference curve modification, based on recommendations for light exposure [69], left: individual with poor circadian light hygiene (DDIbl = 473 NEIbl = 13.38); right: individual with fair light hygiene (DDIbl = 397 NEIbl = 0.17). DDIbl—Daylight Deficit Index for blue light exposure (green areas—area under the curve of recommended daylight); NEIbl—Nocturnal Excess Index of blue light exposure (black areas—area above the curve of recommended nocturnal light).

**Table 1 diagnostics-15-00327-t001:** Wearables for circadian medicine/chronomedicine.

Accelerometry-Based:	Others:
Actigraphs: usually wrist-worn or hip-worn devices that track movement to estimate sleep patterns over extended periods, some can also measure physical activity, skin and ambient temperature, and light exposure. Issues to resolve: lack of unified standards for quality of minimal sampling requirements, light sensors are distant from organ of vision and can be covered by clothes. Poor precision in assessment of sleep latency.	Heart Rate Monitors: can be used as stand-alone devices or integrated into other wearables, offering data on heart rate variability, which can correlate with cardiovascular health, stress and sleep quality. Limitation: highly variable and depends on various external and internal factors.
Sleep Trackers: specialized watches and bands that monitor sleep stages, and overall sleep quality. Merit: can be used more widely than professional devices. Issues include inconsistent and limited accuracy across devices, as well as reduced reliability when compared to professional actigraphs.	Blood Pressure Monitors: Blood pressure monitors can be utilized in chronomedicine to track 24 h changes in blood pressure throughout the day and night and how these changes vary from day to day. Limitation: regarded obtrusive by most users, interfere with quality of sleep.
Smart Watches: commercially available devices that come with built-in features to track physical activity. Merit: can be used more widely than professional devices. Issues include inconsistent and limited accuracy across devices, as well as reduced reliability when compared to professional actigraphs.	Continuous Glucose Monitors (CGMs): while not exclusively for circadian research, CGMs can provide data on metabolic changes throughout the day and night. From the viewpoint of diabetes research, glucose variability is now considered as important a biomarker as the A1c. Sensitive to meal regimens and contents, which can be both limitation and merit depending on purpose of research. However, CGMs are invaluable for health in patients with diabetes.
Fitness Trackers: similar to smartwatches, these devices track steps, activity, heart rate, all of which can be relevant to circadian rhythms. Merit: can be used more widely than professional devices. Issues include inconsistent and limited accuracy across devices, as well as reduced reliability when compared to professional actigraphs.	Photopletysmography (PPG) Monitors: PPG monitors can be used to measure changes in blood volume in the body, providing valuable data on heart rate variability and respiration rate. By analyzing these metrics over a 24 h period, healthcare providers can gain insight into an individual’s circadian rhythm and identify any deviations or abnormalities that may be indicative of increased risk of disease. Benefits: cost-effective, user-friendly, non-invasive, real-time monitoring of cardiovascular metrics. Limitations: sensitivity to motion artifacts, variability in accuracy based on skin characteristics.
Smart Rings: devices like the Oura Ring track sleep, temperature, and activity, and light providing insights into 24 h rhythms. Merits and issues are overall similar to smart watches and fitness trackers. Can be regarded as more convenient by some users. Light sensors are less dependent on clothes, while depend on gloves in location with low ambient temperature.	Wearables for Monitoring Biochemicals: wearables that can monitor biochemicals such as C-reactive protein, interleukin-1b, and cortisol can provide important information on inflammation levels, immune system function, and stress levels throughout the day and night. By tracking these biochemical markers over time, healthcare providers can better understand how individuals’ circadian rhythm may be influencing their overall health and make informed treatment decisions. Merits: offers unique opportunity to monitor biochemicals and gene expression providing unprecedented personalized insight into non-invasive health tracking. Limitations: newly evolving field, which encounters challenges of data accuracy, interpretation, privacy concerns, and the complexities of obtaining necessary approvals and compliance with health regulations.

Footnote: Each accelerometry-based device can contribute to a better understanding of circadian rhythms and their effect on health, though it is important to note that not all are validated for research purposes to the same extent as actigraphy.

**Table 2 diagnostics-15-00327-t002:** Most common forms of circadian disruptions: typical and alternative features.

Circadian Disruption Marker	Typical Deviation	Alternatives/Comments
Circadian rhythm measures		
Amplitude	↓ [11,12,36,38,56,61,62,65,66,67,68,69,70,71,72,73,74,75,76,77,78,79,80,81,82,83]	↑ Transient elevation of amplitude due to jet lag, or shift work, over-swinging functions such as blood pressure [84,85,86,87]
Phase	Delay → [61,72,79,80,81,82,88,89,90,91]	Both delay → or advance ← (optimal phase position is determined by circadian clock precision [81,91,92,93]
Waveform (circadian robustness)	Reduced fitness of the curve to predictable best-fitted model [36,61,62,72]	Flexibility rather than rigidity can be useful for adaptive needs, i.e., heart rate variability
Variability measures		
Fragmentation and regularity (activity and sleep)	IV (intra-daily variability) ↑ IS (inter-daily stability) ↓ [38,51,61,62,72,74,75,77,80,94]	Can be beneficial in certain cases such as short-term adaptation requiring high vigilance [95,96] Large inter- and intra-individual differences in the duration of sleep cycles throughout the night [46,47]
Spectral composition	Extra-circadian dissemination (ECD): ratio between circadian and non-circadian (ultradian/infradian) amplitudes ↓ [2,10,11,12,97]	Some ultradian and infradian components are built-in and beneficial for health [47,98,99,100,101,102]
Alignment	Misalignment: disturbed phase relationship between circadian marker rhythms [55,56,103]	Optimal phase angles may vary depending on genotype, age, light environment: season and latitude; and meal timing if peripheral rhythms’ phases are considered.
Composite markers (area under curve), AUCs; function-on-scalar regression (FOSR), etc.	↑ or ↓ [53,54,63,76,104,105,106]	Optimal reference curves may depend on genotype, age, light environment: season and latitude; and meal timing if peripheral rhythms’ phases are considered.
Social jet lag	↑ [107,108,109,110,111,112,113,114]	Largely varies with age [111] and social obligations [115], may depend on the number of actual working days per week. All these factors can modify hazards of SJL for health.

**Table 3 diagnostics-15-00327-t003:** Summary of methods for analyzing wearable device data.

Method	Advantages	Disadvantages
Moving Linear Regression Models	Easy to implement and interpret, widely accessible. Good agreement with questionnaires [148].	Assume linear relationships.
Non-parametric Actigraphy Indices	Effectively capture informative features of the circadian rhythms of activity and light exposure such as their extent of irregularity; less sensitive to noise in the data.	Provide information on some features of the rhythm but do not quantify the shape of circadian rhythms. Interpretation can be subjective, when sleep patterns are irregular. Non-parametric analyses might overlook underlying multi-frequency patterns or relationships that parametric methods capture.
Parametric Cosinor Analysis	Effectively models periodic data and quantifies amplitude and acrophase. Optimal for oscillations that demonstrate a tied fit to the fitted model. Provides statistical tests for the presence of a rhythm and a measure of uncertainty for the estimation of its parameters, making it easier to draw conclusions about the presence and shape of rhythms.	May require the consideration of multiple harmonic terms for non-sinusoidal waveforms.
Approximation-Based Least-Squares Methods	Flexible for fitting various non-linear models. Can be applied to a wide range of models, including linear, polynomial, and non-linear functions. Effective in minimizing residuals, thus enabling a better model fit.	Can become complex to implement and interpret with intricate models. Require advanced computational resources. Higher-order modeling overestimates amplitude when data have gaps [138].
Fourier Transformation	Provides analysis of periodic components, which is effective for analysis of periodic signals to identify dominant frequencies. Transformation of time-domain data into frequency-domain amplitudes and phases at specified frequencies simplifies the identification of cycles.	Assumption that the signal is stationary (its statistical properties do not change over time) is not always true for real-life data. Provides complex output, which may lack physiological relevance and requires additional expertise for meaningful interpretation of the results.
Wavelet Transformation	Decomposes a signal into wavelets, localized in both time and frequency. This method is useful for analyzing non-stationary signals. Provides information on the dominant mode of variability and how it varies over time.	Requirement of choosing the appropriate wavelet for a specific application is often not straightforward and may require trial and error, and may often lack physiological relevance. Aimed to analyze non-stationary signals, it relies on certain assumptions of signal behavior that may not hold true for physiological data from wearables, potentially leading to inaccurate conclusions.
Hilbert Transformation	Useful to analyze the phase of the reference signal against the phase of the target signal and measure phase relationships between signals [147]. Can be used to decipher non-linear and non-stationary signals.	While it can be applied for non-stationary signals, this method assumes that the signal is relatively smooth and continuous. Swift changes in the amplitude-phase domain, data gaps and noisy data limit the accuracy of the results
Artificial Neural Networks	Capable of modeling complex, non-linear relationships. Can learn from large datasets and improve over time.	Requires large databases for training. Often seen as a “black box,” making the interpretation of results challenging and less transparent.
Limit-Cycle Oscillator Models	Provide a dynamic representation of circadian rhythms. Can incorporate feedback mechanisms, adding depth to the modeling process.	Complex to implement and understand, which may deter some users. Require advanced computational resources and expertise, making them less accessible.

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
