# Peer review of "Wearables in Chronomedicine and Interpretation of Circadian Health"

_diagnostics, 2025, doi:10.3390/diagnostics15030327_

Round 1

Reviewer 1 Report

Comments and Suggestions for Authors

This is the first review of chronomedicine I’ve read that I can recall specifically highlighting amplitude, and I thank the authors for calling attention to it. That said, I think this review has some structural issues that make it hard to read, and a critical piece of the literature on wearables and chronomedicine is missing (limit cycle oscillator models for predicting circadian phase). 

To speak to the second point first: I was expecting to see references to papers like 

  • Stone et al. 2019 (“Application of a Limit-Cycle Oscillator Model for Prediction of Circadian Phase in Rotating Night Shift Workers”), 

  • Huang et al 2021 (“Predicting circadian phase across populations: a comparison of mathematical models and wearable devices”), and 

  • Cheng et al 2021 (“Predicting circadian misalignment with wearable technology: validation of wrist-worn actigraphy and photometry in night shift workers”).

These papers represent one of the most exciting areas of active research in this space right now—using actigraphy and limit cycle oscillator models to predict gold standard biomarkers of the clock, like dim light melatonin onset, as well as health outcomes. These approaches are in use by research groups all around the world (e.g. Lim et al. 2024, “Accurately predicting mood episodes in mood disorder patients using wearable sleep and circadian rhythm features”), as well as in the U.K. Biobank analysis (Windred et al. 2024 “Brighter nights and darker days predict higher mortality risk: A prospective analysis of personal light exposure in >88,000 individuals”). They also are far more robust to outliers than cosinor analysis and much more flexible than measures like IV, IS, and SJL. Without these approaches heavily featured, this review will not be an accurate snapshot of the field in this moment. 

My second piece of feedback is that the review needs to be rewritten to improve clarity. 

Here are some of the things I struggled with: 

  • Very long paragraphs (e.g., pages 2, 9, 12)

  • Bulleted lists in the middle of paragraphs (i.e., in the two section 3 headers)

  • No spacing in between columns in the tables

  • Blurry figures with very small font 

  • Sections that don’t seem to be released to wearables; e.g. Molecular Insights on Interaction between Timed Physical Activity and Brain Health

  • Repeated sentences; e.g. “In the field of circadian medicine research, and chronomedicine more generally…”

  • Terms being used before they were defined.

Here are some specific requests of revisions as examples of what I’m looking for: 

What is circadian medicine? What is chronomedicine? The authors imply circadian medicine is a subset of chronomedicine, but I’ve actually heard it the opposite way: circadian medicine is all circadian-related health (e.g., light and mood), and chronomedicine is specifically timing drugs for optimal circadian effect. It’s fine if the authors have different definitions, but they should explicitly include them. 

In the introduction, the authors cover a lot and bring up parametric and nonparametric without defining them. I think they should cut back the introduction quite a bit, and focus on areas that are not covered elsewhere and make sense in an introduction; like defining key components of a rhythm (phase, amplitude, period) and how they are traditionally measured.

I had a hard time following the logical flows in some places. For instance, I struggled with understanding what lines 97-102 were trying to say (my interpretations in parentheses):

  • “Overall, modern wearable monitoring techniques offer unprecedented perspectives to define normal ranges for rhythm parameters and use them to track unspecific and disease-specific health risks.” (Wearables let us track things we couldn’t track before. We can use this to come up with healthy ranges.)
  • “It is advisable for innovative devices to be aimed specifically at certain patient groups.” (It’s best for new devices to be used within specific patient groups (?)) 

  • “For instance, the association between actigraphy and single-channel EEG was found to be weakened in individuals with mild cognitive impairments and those diagnosed with Alzheimer's disease.” 

What I think the authors are saying is that the healthy ranges we can identify in the public at large are not likely to be the same in patient populations, and we should be aware of this. But that’s not coming through in the sentence “It is advisable for innovative devices to be aimed specifically at certain patient groups,” and I’m also not sure why this needs more than a sentence in a review of wearables in chronomedicine.

In sum, while I think a review of this kind has value, this review as written leaves out a lot of work by leading groups in this space (KAIST, Monash, Michigan/Henry Ford, Harvard) and needs significant edits for clarity.

Comments on the Quality of English Language

See above

Author Response

This is the first review of chronomedicine I’ve read that I can recall specifically highlighting amplitude, and I thank the authors for calling attention to it. That said, I think this review has some structural issues that make it hard to read, and a critical piece of the literature on wearables and chronomedicine is missing (limit cycle oscillator models for predicting circadian phase).

Q1 To speak to the second point first: I was expecting to see references to papers like

Stone et al. 2019 (“Application of a Limit-Cycle Oscillator Model for Prediction of Circadian Phase in Rotating Night Shift Workers”),

Huang et al 2021 (“Predicting circadian phase across populations: a comparison of mathematical models and wearable devices”), and

Cheng et al 2021 (“Predicting circadian misalignment with wearable technology: validation of wrist-worn actigraphy and photometry in night shift workers”).

These papers represent one of the most exciting areas of active research in this space right now—using actigraphy and limit cycle oscillator models to predict gold standard biomarkers of the clock, like dim light melatonin onset, as well as health outcomes. These approaches are in use by research groups all around the world (e.g. Lim et al. 2024, “Accurately predicting mood episodes in mood disorder patients using wearable sleep and circadian rhythm features”), as well as in the U.K. Biobank analysis (Windred et al. 2024 “Brighter nights and darker days predict higher mortality risk: A prospective analysis of personal light exposure in >88,000 individuals”). They also are far more robust to outliers than cosinor analysis and much more flexible than measures like IV, IS, and SJL. Without these approaches heavily featured, this review will not be an accurate snapshot of the field in this moment.

R1 We sincerely appreciate the reviewer for thorough evaluation of our work and for raising constructive questions. We acknowledge that the limit-cycle oscillator model (as well as the other models, targeted on amplitude-phase precision assessment/prediction) is a valuable approach, particularly for shift workers and in cases where activity data lacks accompanying information on light exposure. This model is especially relevant in sleep research under challenging conditions when phase is largely unstable. As we mentioned at the beginning of our review, our primary focus was to explore approaches that extend beyond sleep research and investigate other physiological rhythms that are less studied. Consequently, topics such as sleep phase and shift work fell outside the main scope of our review.

However, we fully agree that incorporating the limit-cycle oscillator model, as well as Neural Network Modeling, Approximation-Based Least-Squares Method, Signal Transformation modeling and also the sleep variability index) into our discussion would enhance the overall quality of the review. Therefore, we added a new Table to provide relative comparison of pros and cons of different methods, applied to analyze data from different wearable devices.

We now added a new Table 3: Summary of Methods for Analyzing Wearable Device Data and the following text:

Different approaches for the precise assessment of circadian phase in field studies can be used with relative merits that may depend on research purposes and conditions (Stone et al., 2020; Kim et al., 2020), and also on the extent of alignment between activity and light exposure (Brown L et al., 2021). For certain purposes, such as delayed sleep–wake phase disorder, light-based methods incorporating multiple linear regression of light exposure for phase assessment were recommended (Murray et al., 2021), while activity-based models can be useful in older adults (aged 58 to 86 years) (Mayer et al., 2024) who tend to advance their sleep phase. Activity-based phase modeling has been shown to outperform light-based modeling in predicting the dim light melatonin onset (DLMO) in shift workers (Huang et al., 2021). Our own data suggest that the same may hold true for cosinor-based models. However, their effectiveness may depend on ambient light conditions and the circadian light hygiene index (Gubin et al., 2024romj), both of which can vary significantly with the seasons (Gubin et al., 2025a). Furthermore, in view of substantial individual differences in light sensitivity (Phillips et al., 2019), predictions can vary widely among individuals, influenced not only by genetic factors but also by other co-factors, such as a history of COVID-19. This history can affect circadian patterns and light sensitivity related to actigraphy-based measures (Gubin et al., 2024d; Gubin et al., 2025b) as well as blood pressure (Shurkevich et al., 2024).

Estimates of amplitude and phase as proxies of circadian rhythm may require different methods with distinct inherent strengths in modelling biological oscillations:

Cosinor Analysis was developed specifically for the purpose of modelling cyclic data and is particularly useful for estimating rhythm parameters like amplitude and acrophase (Cornelissen, 2014; Mitchell et al., 2017; Hirten et al., 2021; Doyle et al., 2022; Gombert-Labedens et al., 2024; Shim et al., 2024; Gubin et al., 2024, 2025b). While the single-component model assumes the pattern to be sinusoidal, consideration of additional harmonic terms with periods of e.g., 12, 8, and 6 h can approximate the waveform more precisely, including phase estimation.

Non-parametric Actigraphy Indices capture informative features of the circadian rhythms of activity and light exposure such as their extent of irregularity. However, their precision in quantifying amplitude may be more subjective in interpretation, leading to greater variability in results.

Moving Linear Regression Models used for sleep/wake scoring over short (e.g., 1-minute) intervals are widely applied to characterize basic sleep parameters (33 Patterson; Cole et al., 1992; Sazonov et al., 2004), thereby capturing important features of the circadian rhythm other than the amplitude and phase.

Artificial Neural Networks (ANN) are capable of handling complex correlations and can therefore determine amplitude and phase for various types of data (Kolodyazhniy et al., 2012; Stone et al., 2019; Dijk & Duffy, 2020). Nevertheless, the efficiency of the networks directly depends on the availability of sufficient data for training. These techniques may not be as easy to use for the purpose of estimating amplitude and phase as is the case with cosinor. Results from a study using actigraphy data of blue light and temperature indicated that the ANN model was capable of predicting circadian timing within ± 2 hours for most individuals following diurnal schedules. This method, however, did not extend to night shift scenarios (Stone et al., 2019).

Limit-Cycle Oscillator Modeling effectively represents biological oscillations and can provide information on feedback mechanisms, allowing dynamic estimations of amplitude and phase. On the other hand, their complexity and need for extensive computational resources can limit their practical applications. Limit-Cycle Oscillator Modeling is most effective for estimating parameters with non-sinusoidal patterns and when phase stability is low (shift-work) (Stone et al., 2019; Cheng et al., 2021).

Approximation-Based Least-Squares Methods (Kim et al., 2023) offer significant versatility, being capable of accommodating a wide range of models, including non-linear ones, which makes it particularly useful for estimating amplitude and phase. These methods efficiently reduce residuals to enhance model fit; however, they can require substantial computational resources, especially when working with more complex models. Additionally, their implementation may be less convenient and more challenging to interpret, potentially complicating their practical application.

Furthermore, several methods can improve cosine decomposition, particularly in investigations of ultradian oscillations and their interpretation:

Fourier transformation decomposes a signal into constituent frequencies, allowing for the analysis of periodic components. It is effective for analyzing periodic signals and identifying dominant frequencies in spectra (Neikrug et al., 2020). Fourier transformation can utilize walking accelerometer data to predict somatic health (Werkmann et al., 2024), and can outperform a non-parametric approach in identifying depression (George et al., 2021). However, like other methods, it can be sensitive to noise, outliers, and a non-stationarities of the signal (when mean, variance, and autocorrelation structure are not consistent over time).

Wavelet transformation (Vuong et al., 2023; Ahmed et al., 2024) allows a thorough analysis of non-stationary signals and detect transient features in circadian and extra-circadian signals. However, it is more complex to implement and interpret than traditional methods and requires careful and subjective selection of parameters.

Hilbert transformation (Lin & Zhu, 2012; Xu et al., 2016; Zschocke et al., 2019) is useful for extracting instantaneous frequency and amplitude of oscillations, providing insights into complex rhythmic patterns and their phase relationship (Chen et al., 2024). It can be used for analysing non-linear and non-stationary signals. However, it is also sensitive to noise and requires continuous data, furthermore interpretation may be challenging without a solid understanding of the underlying mathematics. The choice between different methods of actigraphy/wearables data analyses largely depends on the specific research goals, the sources of the data, and the expected rhythms to be analyzed.

Table 3 provides a brief comparison of the relative merits of the different methods used to analyze data from wearables.

Disadvantages

Advantages

Method

Assume linear relationships.

Easy to implement and interpret, widely accessible.

Good agreement with questionnaires [357].

Moving Linear Regression Models

Provide information on some features of the rhythm but do not quantify the shape of circadian rhythms. Interpretation can be subjective, when sleep patterns are irregular. Non-parametric analyses might overlook underlying multi-frequency patterns or relationships that parametric methods capture.

Effectively capture informative features of the circadian rhythms of activity and light exposure such as their extent of irregularity; less sensitive to noise in the data.

Non-parametric Actigraphy Indices

May require the consideration of multiple harmonic terms for non-sinusoidal waveforms.

Effectively models periodic data and quantifies amplitude and acrophase. Optimal for oscillations that demonstrate a tied fit to the fitted model. Provides statistical tests for the presence of a rhythm and a measure of uncertainty for the estimation of its parameters, making it easier to draw conclusions about the presence and shape of rhythms.

Parametric Cosinor Analysis

Can become complex to implement and interpret with intricate models.

Require advanced computational resources. Higher-order modeling overestimates amplitude when data have gaps [93].

Flexible for fitting various non-linear models. Can be applied to a wide range of models, including linear, polynomial, and nonlinear functions.

Effective in minimizing residuals, thus enabling a better model fit.

Approximation-Based Least-Squares Methods

Assumption that the signal is stationary (its statistical properties do not change over time) is not always true for real-life data.

Provides complex output, which may lack physiological relevance and requires additional expertise for meaningful interpretation of the results.

Provides analysis of periodic components, which is effective for analysis of periodic signals to identify dominant frequencies.

Transformation of time-domain data into frequency-domain amplitudes and phases at specified frequencies simplifies the identification of cycles.

Fourier Transformation

Requirement of choosing the appropriate wavelet for a specific application is often not straightforward and may require trial and error, and may often lack physiological relevance.

Aimed to analyze non-stationary signals, it relies on certain assumptions of signal behavior that may not hold true for physiological data from wearables, potentially leading to inaccurate conclusions.

Decomposes a signal into wavelets, localized in both time and frequency. This method is useful for analyzing non-stationary signals.

Provides information on the dominant mode of variability and how it varies over time.

Wavelet Transformation

While it can be applied for non-stationary signals, this method assumes that the signal is relatively smooth and continuous.

Swift changes in the amplitude-phase domain, data gaps and noisy data limit the accuracy of the results

Useful to analyze the phase of the reference signal against the phase of the target signal and measure phase relationships between signals [102].

Can be used to decipher non-linear and non-stationary signals.

Hilbert Transformation

Requires large databases for training.

Often seen as a "black box," making the interpretation of results challenging and less transparent.

Capable of modeling complex, non-linear relationships.

Can learn from large datasets and improve over time.

Artificial Neural Networks

Complex to implement and understand, which may deter some users.

Require advanced computational resources and expertise, making them less accessible.

Provide a dynamic representation of circadian rhythms.

Can incorporate feedback mechanisms, adding depth to the modeling process.

Limit-Cycle Oscillator Models

Q2 My second piece of feedback is that the review needs to be rewritten to improve clarity.

Here are some of the things I struggled with:

Very long paragraphs (e.g., pages 2, 9, 12)

Bulleted lists in the middle of paragraphs (i.e., in the two section 3 headers)

R2 We have now omitted bulleted lists and rearranged several paragraphs and the whole sections as also suggested by Reviewer 3.

Q3 No spacing in between columns in the tables

R3 Is now corrected as suggested.

R4 Blurry figures with very small font

A4 All figures are now enhanced in quality and larger fonts are now used.

R5 Sections that don’t seem to be released to wearables; e.g. Molecular Insights on Interaction between Timed Physical Activity and Brain Health

A5 We believe this section enhances the understanding of the complex interplay between changes in rhythmic oscillations, as captured by wearables, along with their causes and consequences.

R6 Repeated sentences; e.g. “In the field of circadian medicine research, and chronomedicine more generally…”

A6 We have clarified our understanding of these definitions upon their initial mention and have avoided repetitions in the subsequent text.

R7 Terms being used before they were defined.

A7 We have now reviewed the text to ensure that definitions for specific terms and abbreviations are provided upon their first mention.

Here are some specific requests of revisions as examples of what I’m looking for:

R8 What is circadian medicine? What is chronomedicine? The authors imply circadian medicine is a subset of chronomedicine, but I’ve actually heard it the opposite way: circadian medicine is all circadian-related health (e.g., light and mood), and chronomedicine is specifically timing drugs for optimal circadian effect. It’s fine if the authors have different definitions, but they should explicitly include them.

A8 As students of the author who introduced to science both the term “circadian” and the definition of “chronomedicine”, we believe that chronomedicine includes aspects broader than “circadian”. By definition, “circadian” deals with rhythms that have a period of about 24-hours. There are rhythms with other periods that have also practical applications. We now added the following text in the first paragraph of section 2: “... this review will also deliberate on the eminent roles of commercially available wearable devices in the sphere of circadian medicine and chronomedicine more broadly, including rhythms with periods outside the circadian domain [1,2,7]

R9 In the introduction, the authors cover a lot and bring up parametric and nonparametric without defining them. I think they should cut back the introduction quite a bit, and focus on areas that are not covered elsewhere and make sense in an introduction; like defining key components of a rhythm (phase, amplitude, period) and how they are traditionally measured.

A9 We now added definitions of MESOR, amplitude and acrophase, and period. We also restricted text in order to comply also with recommendations from the other reviewers.

R10 I had a hard time following the logical flows in some places. For instance, I struggled with understanding what lines 97-102 were trying to say (my interpretations in parentheses):

Overall, modern wearable monitoring techniques offer unprecedented perspectives to define normal ranges for rhythm parameters and use them to track unspecific and disease-specific health risks.” (Wearables let us track things we couldn’t track before. We can use this to come up with healthy ranges.)

A10 We now simplified this sentence as follows: “Overall, innovative wearables and analytical tools work together to derive refined reference ranges and identify new biomarkers to assess health quality and disease risk”.

R11 “It is advisable for innovative devices to be aimed specifically at certain patient groups.” (It’s best for new devices to be used within specific patient groups (?))

A11 We rephrased the text as follows: “New devices should be designed for specific patient groups, also be merit to have new devices for everyone to use”.

R12 “For instance, the association between actigraphy and single-channel EEG was found to be weakened in individuals with mild cognitive impairments and those diagnosed with Alzheimer's disease.” What I think the authors are saying is that the healthy ranges we can identify in the public at large are not likely to be the same in patient populations, and we should be aware of this. But that’s not coming through in the sentence “It is advisable for innovative devices to be aimed specifically at certain patient groups,” and I’m also not sure why this needs more than a sentence in a review of wearables in chronomedicine.

A12 The whole sentence was now rephrased as follows: “It has been emphasized that devices should be evaluated in relevant populations for their intended use [26]”.

R13 In sum, while I think a review of this kind has value, this review as written leaves out a lot of work by leading groups in this space (KAIST, Monash, Michigan/Henry Ford, Harvard) and needs significant edits for clarity.

A13 We are grateful to the reviewer for a thorough evaluation of our work and for raising important questions. Multiple citations to works by each of the mentioned groups were already provided in our initial version of the manuscript and more citations have been added in this revision.

Reviewer 2 Report

Comments and Suggestions for Authors

Clarity of writing

l  The language of the article is relatively precise, with accurate use of technical terminology and overall fluency of expression

l  Some paragraphs of the article are lengthy, especially those dense with technical descriptions

Organization

l  Slightly stiff transitions between some paragraphs, especially in the discussion of circadian and non-circadian rhythms, lack of logical articulation

Innovation

l  For the future trends approach, it is recommended to add the prospect of exploring AI, big data and new sensor technologies

Recommendations for improvement:

1、    It is recommended to add serial numbers after each formula to improve clarity and ease of reference throughout the text.s

2、    The serial number for the titles "3. Interpretation of Deviant Parameters" and "3. Overview of Actigraphic Health Markers" is repeated. Please revise to ensure unique numbering for all titles.

3、    The serial numbers for the subtitles "4.1. Circadian Markers of Morbidity and Mortality" and "4.1. Circadian Markers of Metabolic Health" are duplicated. It is suggested to correct the numbering for consistency.

4、    Section 3, "Overview of Actigraphic Health Markers", lacks empirical evidence or case studies. To enhance its credibility, it is recommended to incorporate specific case studies or empirical data that demonstrate the effectiveness of actigraphic health markers in clinical or personal health monitoring.

5、    The descriptions of parametric and non-parametric indices in Section 3 could be summarized in a table format. This will make the information more concise, visually clear, and easier for readers to compare and understand.

6、    The left portion of Figure 1 requires clearer representation, as the current depiction appears overly scribbled and ambiguous. The right part of the figure needs better alignment and organization to enhance its visual clarity and overall presentation quality.

7、    For Tabel 2 : It is recommended to improve the table's layout for better alignment, add specific details such as monitoring frequency and health applications, and provide a comparison of advantages, limitations, and applicability of the devices to enhance clarity and usability for readers.

8、    For section 4: The section contains an excessive listing of cases, with insufficient correlation between the sub-paragraphs. The general data support provided lacks depth, as specific statistical data are not effectively presented.

9、    For section 5: The studies discussed in this section are presented in a fragmented manner, lacking an overarching systematic approach.

10、For section 7The section provides a discussion on wearable devices for tracking circadian rhythm markers in neurodegenerative diseases. However, there is a lack of comparative analysis regarding the advantages and limitations of different device types, such as GPS, accelerometers, and light sensors.

Author Response

Clarity of writing

Q1 l The language of the article is relatively precise, with accurate use of technical terminology and overall fluency of expression

R1 We are grateful to reviewer for thorough evaluation of our work and raising important questions.

Q2 l Some paragraphs of the article are lengthy, especially those dense with technical descriptions

R2 We have now restructured the text, and rearranged several paragraphs and all sections, as also suggested by other reviewers.

Organization

Q3 l Slightly stiff transitions between some paragraphs, especially in the discussion of circadian and non-circadian rhythms, lack of logical articulation

R3 Discussion section was considerably reorganized, as also suggested by Reviewer 3.

Innovation

Q4 l For the future trends approach, it is recommended to add the prospect of exploring AI, big data and new sensor technologies

R4 Thank you for the suggestion. To address this point, we now added the following concluding paragraph:

New sensor technologies, enabling the collection of big data sets to be organized in large repositories, integrated with artificial intelligence (AI) are expected to further enhance the prediction of health hazards and upgrade the interpretation of circadian health by wearables. Monitoring a broader range of health proxies with standardized sampling requirements, empowered by sophisticated AI algorithms will allow personalized chronodiagnosis and may also encourage large-scale studies to explore the interplay between genetic background, environmental factors, lifestyle factors, and circadian health outcomes. This holistic approach should facilitate the development of targeted interventions to optimize health and well-being in diverse populations, based on their distinct genetic predispositions, and specifics of environmental factors and lifestyle. As these technologies evolve, they are expected to offer truly personalized prevention and management capabilities of circadian-related disorders.”

Recommendations for improvement:

Q5 1It is recommended to add serial numbers after each formula to improve clarity and ease of reference throughout the text.

R5 Done.

Q6 2The serial number for the titles "3. Interpretation of Deviant Parameters" and "3. Overview of Actigraphic Health Markers" is repeated. Please revise to ensure unique numbering for all titles.

R6 Thank you for noting. Corrected.

Q7 3The serial numbers for the subtitles "4.1. Circadian Markers of Morbidity and Mortality" and "4.1. Circadian Markers of Metabolic Health" are duplicated. It is suggested to correct the numbering for consistency.

R7 Thank you for noting. Corrected.

Q8 4Section 3, "Overview of Actigraphic Health Markers", lacks empirical evidence or case studies. To enhance its credibility, it is recommended to incorporate specific case studies or empirical data that demonstrate the effectiveness of actigraphic health markers in clinical or personal health monitoring.

R8 This section aims to provide background information on actigraphic health markers. A broader discussion of their application in various studies will be presented in the following sections. However, already in this section it is underlined that “Most common forms of circadian disruptions that are revealed by actigraphy are summarized in Table 2 and schematized in Figure 1.”

Q9 5The descriptions of parametric and non-parametric indices in Section 3 could be summarized in a table format. This will make the information more concise, visually clear, and easier for readers to compare and understand.

R9 Based on suggestions from Reviewer 1, we now added a Table, which includes different methods of data analysis and interpretation with their relative advantages and disadvantages.

Q10 6The left portion of Figure 1 requires clearer representation, as the current depiction appears overly scribbled and ambiguous. The right part of the figure needs better alignment and organization to enhance its visual clarity and overall presentation quality.

R10 Figure 1 was enhanced in quality, overall fidelity and organization, alignment was improved, explanations were added.

Q11 7For Tabel 2 : It is recommended to improve the table's layout for better alignment, add specific details such as monitoring frequency and health applications, and provide a comparison of advantages, limitations, and applicability of the devices to enhance clarity and usability for readers.

R11 Table 2 presentation is now improved, as also suggested by Reviewer 1.

As there is currently no consensus on standardized monitoring duration and unified requirements for permissible data gaps, previous research varies in these aspects. We referenced the most common monitoring frequencies to date and emphasized the need for standardization in these areas in the main text.

We also added relative advantages and limitations of the mentioned devices in Table 2.

Q12 8For section 4: The section contains an excessive listing of cases, with insufficient correlation between the sub-paragraphs. The general data support provided lacks depth, as specific statistical data are not effectively presented.

R12 We now removed subsections and reorganized materials in a single section “Circadian Health Markers from Actigraphy”.

Q13 9For section 5: The studies discussed in this section are presented in a fragmented manner, lacking an overarching systematic approach.

R13 In line with suggestions from the other reviewers, this section was re-organized, re-positioned and re-titled. It is now Section 8, “Boosting Brain, Vascular and Metabolic Health by Clock Enhancing Strategies”, which has subsections 8.1. Scheduled Physical Activity; 8.2. Light Hygiene and Chronobiotics; and 8.3. Optimizing Weekly Schedules.

Q14 10For section 7The section provides a discussion on wearable devices for tracking circadian rhythm markers in neurodegenerative diseases. However, there is a lack of comparative analysis regarding the advantages and limitations of different device types, such as GPS, accelerometers, and light sensors.

R14 Thank you for your comment. A comparative analysis of the advantages and limitations of various device types, including GPS, accelerometers, and light sensors, warrants dedicated attention in a separate review, as it falls outside the scope of this review. Note that we have previously conducted a comparative analysis of wearables equipped with light sensors, https://doi.org/10.3390/app122211794. We also added the following text in the Section 2: Several important topics, including the comparative analysis of circuit design, power supply, device longevity, measurement accuracy, noise characteristics, calibration methods, and the advantages and limitations of device types such as GPS, accelerometers, and light sensors, as well as the performance and reliability of wearable devices, warrant specific attention in a separate review and are outside the scope of the current review.

Reviewer 3 Report

Comments and Suggestions for Authors

The authors reviewed an important topic by exploring the role of actigraphy and wearable devices in chronomedicine, as well as the mathematical methods for analyzing physiological rhythms. The manuscript offers a valuable discussion. However, there are several concerns regarding the manuscript to improve the clarity, coherence, and scientific rigor.

The comparison between actigraphy and commercial wearable devices could be expanded. The statement that actigraphy is "the scientifically best-validated tool for research in chronomedicine" (lines 105–107) does not fully acknowledge the advanced capabilities of commercial wearables. Actigraphy devices support wireless communication protocols, including Bluetooth, Bluetooth Low Energy (BLE), Wi-Fi, Zigbee, and others. Each of these protocols has unique characteristics in terms of communication range, frequency bands, and power consumption, which are not discussed in the manuscript. Furthermore, while the authors focus on testing methods, they omit key hardware factors that are crucial for a comprehensive comparison. We suggest the authors discuss the influences of circuit design, power supply, device longevity, measurement accuracy, noise characteristics, and calibration methods on the performance and reliability of both actigraphy and wearable devices.

The manuscript’s discussion of mathematical methods for chronomedicine research could benefit from more clarity and depth. The distinction between parametric and non-parametric methods (lines 57–73) is not clearly defined. While parametric methods are described as relying on sinusoidal or multicomponent models, and non-parametric methods are characterized by their avoidance of such models, the distinction between these approaches become unclear in the provided examples (lines 139–141). A more precise definition and clearer categorization of these methods would help readers understand the distinctions more clearly.

In terms of mathematical analysis approach in chronomedicine, the manuscript focused too much on the cosine decomposition. This limits the scope of the manuscript. Contemporary researches increasingly use a wide range of other available techniques, including Fourier transforms, correlation analysis, wavelet transforms, and Hilbert transforms for spectral analysis and rhythm detection. Furthermore, the machine learning and deep learning approaches developed for analyzing multimodal data collected from wearables have also gained significant traction in recent years. These techniques allow for more comprehensive analysis of physiological signals. Incorporating these approaches would broaden the manuscript’s scope and align it more closely with current advancements in the field.

The manuscript also has issues with formatting and the presentation of formulas. For instance, the formulas in lines 187–194 and 201–217 are poorly formatted, and critical parameters like IV and IS are not sufficiently explained. Metrics like the balance index (ABI), transition probability (TP), and self-similarity parameter (α) are mentioned but lack clear mathematical definitions or explanations. Since these metrics have explicit mathematical representations, further details are necessary to ensure readers can understand their significance correctly. Additionally, if these metrics are categorized as non-parametric measures, the justification for this classification should be made clearer.

The manuscript should also be organized according to a more fluent logical flow. Sections 6 and 7, which discuss molecular insights into temporal physical activities and biomarkers in neurodegenerative diseases, respectively, overlap conceptually. Subsection 6.1, which discusses weekly exercise recommendations, seems more like a practical conclusion derived from molecular insights and might be better placed in section conclusion. Furthermore, Section 7 focuses primarily on neurodegenerative diseases but does not address the broader implications of temporal physical activities for other conditions, such as cardiovascular or endocrine diseases. Expanding this discussion would provide a more comprehensive overview. A clearer separation between molecular mechanisms and pathological implications would also improve the manuscript’s readability and coherence.

The References section could also be improved. For example, the statement that ‘actigraphy and EEG recordings in patients with mild cognitive impairment and Alzheimer’s disease show reduced activity’ could be clarified by specifying the brain regions where these changes occur. Providing this detail would help readers understand the implications of the findings and account for potential variability in EEG activity across different brain regions.

Regarding the "Interpretation of Deviant Parameters" section, several points warrant reconsideration. Specifically, the purpose of lines 225-246 is unclear, and their relationship to the subsequent text is ambiguous, making them seem misplaced. Furthermore, the meanings of certain subheadings, such as "3.2. Phase Deviations" and "3.4. Misalignment (Intrinsic Desynchrony)," are not sufficiently clear.

Part 8 lacks clarity, and readers cannot easily discern the content from the subheadings. I recommend that Part 8 should comprise five distinct sections, each dedicated to a specific theme. These themes include the first section on fNIRS, the second on biochemical aspects, the third on mEDI, the fourth on UVA/UVB, and the fifth on the perspective.

The language throughout the manuscript, particularly in the abstract, could be refined for greater clarity and precision. Simplifying overly complex sentences and ensuring consistent terminology would enhance the manuscript’s accessibility and readability. Additionally, including more figures and well-formatted tables would help illustrate key points and make the findings easier to comprehend.

Author Response

Q1 The authors reviewed an important topic by exploring the role of actigraphy and wearable devices in chronomedicine, as well as the mathematical methods for analyzing physiological rhythms. The manuscript offers a valuable discussion. However, there are several concerns regarding the manuscript to improve the clarity, coherence, and scientific rigor.

R1 We are grateful to this reviewer for the thorough evaluation of our work and raising important questions. We tried to address them carefully point by point.

Q2 The comparison between actigraphy and commercial wearable devices could be expanded. The statement that actigraphy is "the scientifically best-validated tool for research in chronomedicine" (lines 105–107) does not fully acknowledge the advanced capabilities of commercial wearables. Actigraphy devices support wireless communication protocols, including Bluetooth, Bluetooth Low Energy (BLE), Wi-Fi, Zigbee, and others. Each of these protocols has unique characteristics in terms of communication range, frequency bands, and power consumption, which are not discussed in the manuscript. Furthermore, while the authors focus on testing methods, they omit key hardware factors that are crucial for a comprehensive comparison. We suggest the authors discuss the influences of circuit design, power supply, device longevity, measurement accuracy, noise characteristics, and calibration methods on the performance and reliability of both actigraphy and wearable devices.

R2 Thank you for your suggestion. A comparative analysis of the circuit design, power supply, device longevity, measurement accuracy, noise characteristics, and calibration methods on the performance and reliability of both actigraphy and wearable devices deserves specific attention in a separate review, and is well outside the scope of our current review. Please note that recently we provided a comparative analysis of seventeen wearables equipped with light sensors for appearance, dimensions, weight, mounting, battery, sensors, features, communication interface, and software at https://doi.org/10.3390/app122211794. We also added the following text in the Section 2: Several important topics, including the comparative analysis of circuit design, power supply, device longevity, measurement accuracy, noise characteristics, calibration methods, and the advantages and limitations of device types such as GPS, accelerometers, and light sensors, as well as the performance and reliability of wearable devices, warrant specific attention in a separate review and are outside the scope of the current review.

Q3 The manuscript’s discussion of mathematical methods for chronomedicine research could benefit from more clarity and depth. The distinction between parametric and non-parametric methods (lines 57–73) is not clearly defined. While parametric methods are described as relying on sinusoidal or multicomponent models, and non-parametric methods are characterized by their avoidance of such models, the distinction between these approaches become unclear in the provided examples (lines 139–141). A more precise definition and clearer categorization of these methods would help readers understand the distinctions more clearly.

R3 Thank you for this comment. As also suggested by other reviewers, we now updated the scope of methods currently used, beyond a comparison between parametric and non-parametric approaches. We added a comprehensive table to compare these methods with other modeling approaches (Moving Linear regression models, Artificial Neural Networks, Limit-Cycle Oscillator Models, and Approximation-Based Least-Squares Method) for their relative merits and limitations.

Q4 In terms of mathematical analysis approach in chronomedicine, the manuscript focused too much on the cosine decomposition. This limits the scope of the manuscript. Contemporary researches increasingly use a wide range of other available techniques, including Fourier transforms, correlation analysis, wavelet transforms, and Hilbert transforms for spectral analysis and rhythm detection. Furthermore, the machine learning and deep learning approaches developed for analyzing multimodal data collected from wearables have also gained significant traction in recent years. These techniques allow for more comprehensive analysis of physiological signals. Incorporating these approaches would broaden the manuscript’s scope and align it more closely with current advancements in the field.

R4 Thank you for this very constructive comment; we have added the following text:

Furthermore, several methods can improve cosine decomposition, particularly in investigations of ultradian oscillations and their interpretation:

Furthermore, several methods can improve cosine decomposition, particularly in investigations of ultradian oscillations and their interpretation:

Fourier transformation decomposes a signal into constituent frequencies, allowing for the analysis of periodic components. It is effective for analyzing periodic signals and identifying dominant frequencies in spectra (Neikrug et al., 2020). Fourier transformation can utilize walking accelerometer data to predict somatic health (Werkmann et al., 2024), and can outperform a non-parametric approach in identifying depression (George et al., 2021). However, like other methods, it can be sensitive to noise, outliers, and a non-stationarities of the signal (when mean, variance, and autocorrelation structure are not consistent over time).

Wavelet transformation (Vuong et al., 2023; Ahmed et al., 2024) allows a thorough analysis of non-stationary signals and detect transient features in circadian and extra-circadian signals. However, it is more complex to implement and interpret than traditional methods and requires careful and subjective selection of parameters.

Hilbert transformation (Lin & Zhu, 2012; Xu et al., 2016; Zschocke et al., 2019) is useful for extracting instantaneous frequency and amplitude of oscillations, providing insights into complex rhythmic patterns and their phase relationship (Chen et al., 2024). It can be used for analysing non-linear and non-stationary signals. However it is also sensitive to noise and requires continuous data, furthermore interpretation may be challenging without a solid understanding of the underlying mathematics. The choice between different methods of actigraphy/wearables data analyses largely depends on the specific research goals, the sources of the data, and the expected rhythms to be analyzed.

Table 3 provides a brief comparison of the relative merits of the different methods used to analyze data from wearables.

Q5 The manuscript also has issues with formatting and the presentation of formulas. For instance, the formulas in lines 187–194 and 201–217 are poorly formatted, and critical parameters like IV and IS are not sufficiently explained. Metrics like the balance index (ABI), transition probability (TP), and self-similarity parameter (α) are mentioned but lack clear mathematical definitions or explanations. Since these metrics have explicit mathematical representations, further details are necessary to ensure readers can understand their significance correctly. Additionally, if these metrics are categorized as non-parametric measures, the justification for this classification should be made clearer.

R5 Thank you for this notion. To address this issue, we added the following explanation in the text:

IV (Intra-daily Variability) estimates how variable activity is within a day and can range from 0 to ∞, where higher values represent higher fragmentation. IV is calculated by the following formula, where N – total number of measurements in the full time series, Xiindividuals values at time i, Xmean of all Xi values:

IS (Inter-daily Stability) measures how constant the rest-activity pattern is between days and ranges from 0 to 1. Values closer to 1 mean more constant rest-activity patterns. Assuming that measurements are binned over hourly equal intervals per day over the whole time series, IS is calculated by the following formula, where N – total number of measurements in the overall time series, p – number of data per 24 h, Xiindividuals values at time i, Xmean of the overall time series, Xhhourly means:

The Activity balance index (ABI) estimates how balanced activity is during the observation span. It ranges from 0 to 1, with values closer to 1 meaning a more balanced activity distribution.

The Transition probability (TP) estimates transitions from rest to activity state (A→R, AR) or vice versa (R→A) at a given time point t. TP ranges from 0 to 1. Higher values indicate higher transitions from activity to rest or vice versa.

The Self-similarity parameter (α) approaches self-similarity of the acceleration signal during the observation span. It ranges from 0 to 2. If the values fall between 0 and 1, it indicates that the motion is steady and predictable. When the values are between 1 and 2, it suggests that the motion is more erratic and unpredictable. There are some key points to note: a value of 0.5 represents random noise, 1 indicates a pattern called fractal noise, and 1.5 signifies a random walk, which is a type of movement that seems to take random steps.

Q6 The manuscript should also be organized according to a more fluent logical flow. Sections 6 and 7, which discuss molecular insights into temporal physical activities and biomarkers in neurodegenerative diseases, respectively, overlap conceptually. Subsection 6.1, which discusses weekly exercise recommendations, seems more like a practical conclusion derived from molecular insights and might be better placed in section conclusion. Furthermore, Section 7 focuses primarily on neurodegenerative diseases but does not address the broader implications of temporal physical activities for other conditions, such as cardiovascular or endocrine diseases. Expanding this discussion would provide a more comprehensive overview. A clearer separation between molecular mechanisms and pathological implications would also improve the manuscript’s readability and coherence.

R6 According to this recommendation, we now reorganized the sections, moving section 6.1 (Weekly exercise recommendations) in new Section 8. “Boosting Brain, Vascular and Metabolic Health by Clock Enhancing Strategies”, which now includes subsections: 8.1. Scheduled Physical Activity, and 8.2. Light Hygiene and Chronobiotics, and 8.3. Optimizing Weekly Schedules. In this section we also provided a broader overview of implications of temporal physical activities for cardiovascular or endocrine diseases, adding the following text:

For 8.1. Scheduled Physical Activity:

Exercise timing occurring at the post-absorptive phase was shown to cause greater fat oxidation compared to postprandial exercise, as confirmed by indirect calorimetry and 13C magnetic resonance spectroscopy (Iwayama et al., 2023). Timed physical activity such as morning exercise decreased abdominal fat and blood pressure in women, while evening exercise improved muscular performance. In men, evening workouts boosted fat oxidation and lowered systolic blood pressure regardless of macronutrient intake (Arciero et al., 2022). Timed physical activity is promising to handle neurodegenerative pathologies [248].

For 8.2. Light Hygiene and Chronobiotics:

Evening administration of low-dose melatonin lowered high blood pressure in hypertensive patients [236] and intraocular pressure in patients with glaucoma [76], with the greatest lowering effect achieved in the morning. Melatonin also reduced glycosylated hemoglobin levels (HbA1c) and increased high-density lipoprotein-cholesterol (Bazyar et al., 2022). Melatonin can be more effective in combination with enhanced dynamic range of light exposure. ….

Timed physical activity may have greater circadian effects when it coincides with daylight exposure [141], better circadian light hygiene per se may have metabolic benefits (Gubin et al, 2024rmj). At high latitudes, a larger amplitude and an earlier phase of light exposure, mirrored by a greater amplitude and an earlier phase of melatonin and by an earlier sleep phase characterize seasons with a more comfortable circadian light hygiene. These features are also associated with better proxies of metabolic health, even when there are no differences in the patterns of physical activity (Gubin et al., 2025a). Proper timing of intake of some other substances with pronounced circadian effects, such as coffee, can also provide benefits for metabolism and overall health: morning coffee consumption is strongly associated with lower all-cause, cardiovascular, and cancer-specific mortality (Wang et al., 2025). It can also enhance effects of timed exercise by boosting peak power, readiness for physical efforts, and cognitive performance (Duncan et al., 2019).

Q7 The References section could also be improved. For example, the statement that ‘actigraphy and EEG recordings in patients with mild cognitive impairment and Alzheimer’s disease show reduced activity’ could be clarified by specifying the brain regions where these changes occur. Providing this detail would help readers understand the implications of the findings and account for potential variability in EEG activity across different brain regions.

R7 Thank you for this note. Please note, however, that the statement referring to the publication by della Monica et al., (2024) did not claim that “actigraphy and EEG recordings in patients with mild cognitive impairment and Alzheimer’s disease show reduced activity”. It only noted that “the association between actigraphy and single-channel electroencephalography (EEG) was found to be weakened in individuals with mild cognitive impairments and those diagnosed with Alzheimer's disease”. It underlined the need for adjusting devices for “certain patient groups”. We now slightly rephrased this statement for higher clarity: “It has been emphasized that devices should be evaluated in relevant populations for their intended use [26]”.

Q8 Regarding the "Interpretation of Deviant Parameters" section, several points warrant reconsideration. Specifically, the purpose of lines 225-246 is unclear, and their relationship to the subsequent text is ambiguous, making them seem misplaced. Furthermore, the meanings of certain subheadings, such as "3.2. Phase Deviations" and "3.4. Misalignment (Intrinsic Desynchrony)," are not sufficiently clear.

R8 We now rearranged these sections, moving bulleted points to sections with their descriptions and we combined former sections 3.2 and 3.4 into one, which discusses phase deviations – both phase of an individual variable and phase agreement between variables (alignment/misalignment).

Q9 Part 8 lacks clarity, and readers cannot easily discern the content from the subheadings. I recommend that Part 8 should comprise five distinct sections, each dedicated to a specific theme. These themes include the first section on fNIRS, the second on biochemical aspects, the third on mEDI, the fourth on UVA/UVB, and the fifth on the perspective.

R9 We are grateful for this comment. We now restructured former Part 8 (which is now Part 9) into 4 sections: 9.1 fNIRS and PPG; 9.2. Biochemical variables; 9.3. Improving Light Exposure Monitoring and Analytics (in which we combined mEDI and UVA/UVB); and 9.4 Perspectives.

Q10 The language throughout the manuscript, particularly in the abstract, could be refined for greater clarity and precision. Simplifying overly complex sentences and ensuring consistent terminology would enhance the manuscript’s accessibility and readability. Additionally, including more figures and well-formatted tables would help illustrate key points and make the findings easier to comprehend.

R10 We are grateful to this reviewer for the thorough evaluation of our work and raising important questions. We improved the text and Abstract, enhanced figures and tables, and added a new Table on the comparison of advantages and disadvantages of different methods of data analysis.

Round 2

Reviewer 2 Report

Comments and Suggestions for Authors

The authors have responded to all the review comments.